# A Non-stop identity complex (NIC) supervises enterocyte identity and protects from premature aging

Neta Erez[1], Lena Israitel[1], Eliya Bitman-Lotan[1], Wing H Wong[2], Gal Raz[1], Dayanne V Cornelio-Parra[3], Salwa Danial[1], Na'ama Flint Brodsly[1], Elena Belova[4], Oksana Maksimenko[4], Pavel Georgiev[4], Todd Druley[2], Ryan D Mohan[3], Amir Orian[1]*

[1]Rappaport Research Institute and Faculty of Medicine, Technion-Israel Institute of Technology, Haifa, Israel; [2]Division of Pediatric Hematology and Oncology, Washington University, Saint-Louis, United States; [3]Department of Genetics, Developmental & Evolutionary Biology, School of Biological and Chemical Sciences University of Missouri, Kansas City, United States; [4]Department of the Control of Genetic Processes, Institute of Gene Biology Russian Academy of Sciences, Moscow, Russian Federation

**Abstract** A hallmark of aging is loss of differentiated cell identity. Aged *Drosophila* midgut differentiated enterocytes (ECs) lose their identity, impairing tissue homeostasis. To discover identity regulators, we performed an RNAi screen targeting ubiquitin-related genes in ECs. Seventeen genes were identified, including the deubiquitinase Non-stop (CG4166). Lineage tracing established that acute loss of Non-stop in young ECs phenocopies aged ECs at cellular and tissue levels. Proteomic analysis unveiled that Non-stop maintains identity as part of a Non-stop identity complex (NIC) containing E(y)2, Sgf11, Cp190, (Mod) mdg4, and Nup98. Non-stop ensured chromatin accessibility, maintaining the EC-gene signature, and protected NIC subunit stability. Upon aging, the levels of Non-stop and NIC subunits declined, distorting the unique organization of the EC nucleus. Maintaining youthful levels of Non-stop in wildtype aged ECs safeguards NIC subunits, nuclear organization, and suppressed aging phenotypes. Thus, Non-stop and NIC, supervise EC identity and protects from premature aging.

*For correspondence:
mdoryan@tx.technion.ac.il

**Competing interests:** The authors declare that no competing interests exist.

## Introduction

Differentiated cell states are actively established and maintained through action of 'identity supervisors' (*Holmberg and Perlmann, 2012*; *Natoli, 2010*). Identity supervisors safeguard the expression of genes that enable differentiated cells to respond to environmental cues and perform required physiological tasks. Concomitantly, they ensure silencing/repression of previous fate and non-relevant gene programs and reduce transcriptional noise. Inability to safeguard cell identity is a hallmark of aging and results in diseases such as neurodegeneration, diabetes, and cancer (*Bensellam et al., 2018*; *Hudish et al., 2019*; *Conway et al., 2015*; *Hnisz et al., 2013*; *Deneris and Hobert, 2014*). In many cases, transcription factors (TFs) together with chromatin regulators and architectural/scaffold proteins establish and maintain large-scale chromatin and nuclear organization that is unique to the differentiated state of the cell (*Blau and Baltimore, 1991*; *Booth and Brunet, 2016*; *Naetar et al., 2017*; *Bitman-Lotan and Orian, 2018*).

In adult *Drosophila* midgut epithelia, the transcription factor Hey (Hairy/E(spl)-related with YRPW motif), together with *Drosophila* nuclear type A lamin, Lamin C (LamC), co-supervise identity of fully differentiated enterocytes (ECs) (*Monastirioti et al., 2010*; *Gruenbaum and Foisner, 2015*;

*Flint Brodsly et al., 2019*). Highly similar to the vertebrate gut, *Drosophila* midgut epithelia intestinal stem cells (ISC) either self-renew or differentiate into progenitor cells that mature into enteroendocrine cells (EEs) or give rise to enteroblast (EB) progenitors. EBs mature into fully polyploid differentiated enterocytes (ECs) that carry out many critical physiological tasks of the intestine (*Figure 1A*; *Jiang and Edgar, 2012*; *Lemaitre and Miguel-Aliaga, 2013*; *Buchon et al., 2013*; *Hung et al., 2020*). Aging affects the entire midgut, and is associated with loss of EC identity, misdifferentiation of progenitors, pathological activation of the immune system, and loss of the physiological properties of the gut and its integrity. It also results in loss of intestinal compartmentalization, and microbiota-dysbiosis, all leading to reduced lifespan (*Biteau et al., 2010*; *Rera et al., 2012*; *Bonnay et al., 2013*; *Ferrandon, 2013*; *Chen et al., 2014*; *Rodriguez-Fernandez et al., 2020*; *Jasper, 2020*). During aging, the protein levels of identity supervisors such as Hey and LamC decline, resulting in inability to maintain EC-gene programs and ectopic expression of previous- and non-relevant gene programs (*Neves et al., 2015*; *Takeda et al., 2018*; *Flint Brodsly et al., 2019*). Indeed, continuous expression of Hey in aged ECs restores and protects EC identity, gut integrity, and tissue homeostasis (*Flint Brodsly et al., 2019*).

Regulation of EC identity requires signaling to the nucleus to communicate physiological changes in the gut environment. An important mechanism involves changes in post-transcriptional modifications (PTMs) which may propagate, amplify, or conduct signals, ultimately leading to differential gene regulation. One type of PTM is the covalent attachment of ubiquitin or ubiquitin-like (Ub/UbL) molecules, that affect protein stability, function, localization, as well as modulatex chromatin structure (*Swatek and Komander, 2016*; *Heideker and Wertz, 2015*; *Cappadocia and Lima, 2018*; *Song and Luo, 2019*; *Yau et al., 2020*). Recent works suggest an intimate link between ubiquitin, proteostasis, and aging (*Kevei and Hoppe, 2014*; *Vilchez et al., 2014*; *Höhfeld and Hoppe, 2018*; *Enam et al., 2018*; *Chua and Signer, 2020*). We performed an RNAi screen to search for Ub/UbL-related genes within ECs that supervise identity. Screening of 362 genes, 17 were identified whose conditional elimination in fully differentiated ECs resulted in loss of EC identity. Further analysis revealed one of them, the deubiquitinating isopeptidase (DUB) Non-stop (Non-stop/dUSP22) is a key EC identity supervisor. Purification and proteomic analysis identified Non-stop as part of a CP190/Nup98/Sgf11/e(y)2/mdg4 protein complex, termed Non-stop identity complex (NIC), that is essential for maintenance of EC identity. In part, Non-stop protects NIC proteins from age-dependent decline, safeguarding the EC-gene expression signature, as well as large-scale nuclear organization in these cells, preventing premature aging. Over lifespan, Non-stop protein levels in ECs declined, leading to loss of NIC subunits. This decline is associated with loss of gut identity and physiology at the cellular and tissue levels. Maintaining youthful levels of Non-stop prevented loss of the NIC and prevented aging of the gut.

## Results

### A transgenic RNAi screen identified Ub/UbL-related EC identity regulators

To identify EC identity supervisors, a collection of RNAi transgenic flies targeting 362 evolutionarily conserved Ub/UbL-related genes were screened (*Figure 1—source data 2*; List of Ubiquitin-Related Genes according to DRSC - http://www.flyrnai.org/DRSC-SUB.html). Genes were knocked-down in fully differentiated ECs of 2–4 day old adult *Drosophila* females using UAS-RNAi lines and EC-specific conditional driver *MyoIA*-Gal4/Gal80$^{ts}$ coupled system (termed MyoIA$^{ts}$; see methods for specific lines used; *Salmeron et al., 1990*; *Brand and Perrimon, 1993*; *Flint Brodsly et al., 2019*). Conditional RNAi was achieved by shifting flies to 29°C for 48 hr for the indicated time, after which guts were dissected and analyzed. Immunofluorescence was used to score loss of proteins which are hallmarks of EC identity (*Figure 1A,B*). Among the changes upon loss of EC identity is the ectopic expression of the ISC marker Delta on the surface of ECs-like polyploid cells. This change may be also accompanied with a decline in the expression of a GFP signal that is expressed only in fully differentiated ECs (derived from the MyoIA$^{ts}$ Gal4, UAS-GFP transgene). Knockdown of seventeen genes (*Figure 1—source data 2*), resulted in loss of EC identity and appearance of EC-like polyploid cells (PPCs). We also determined whether these genes are required for maintaining identity of

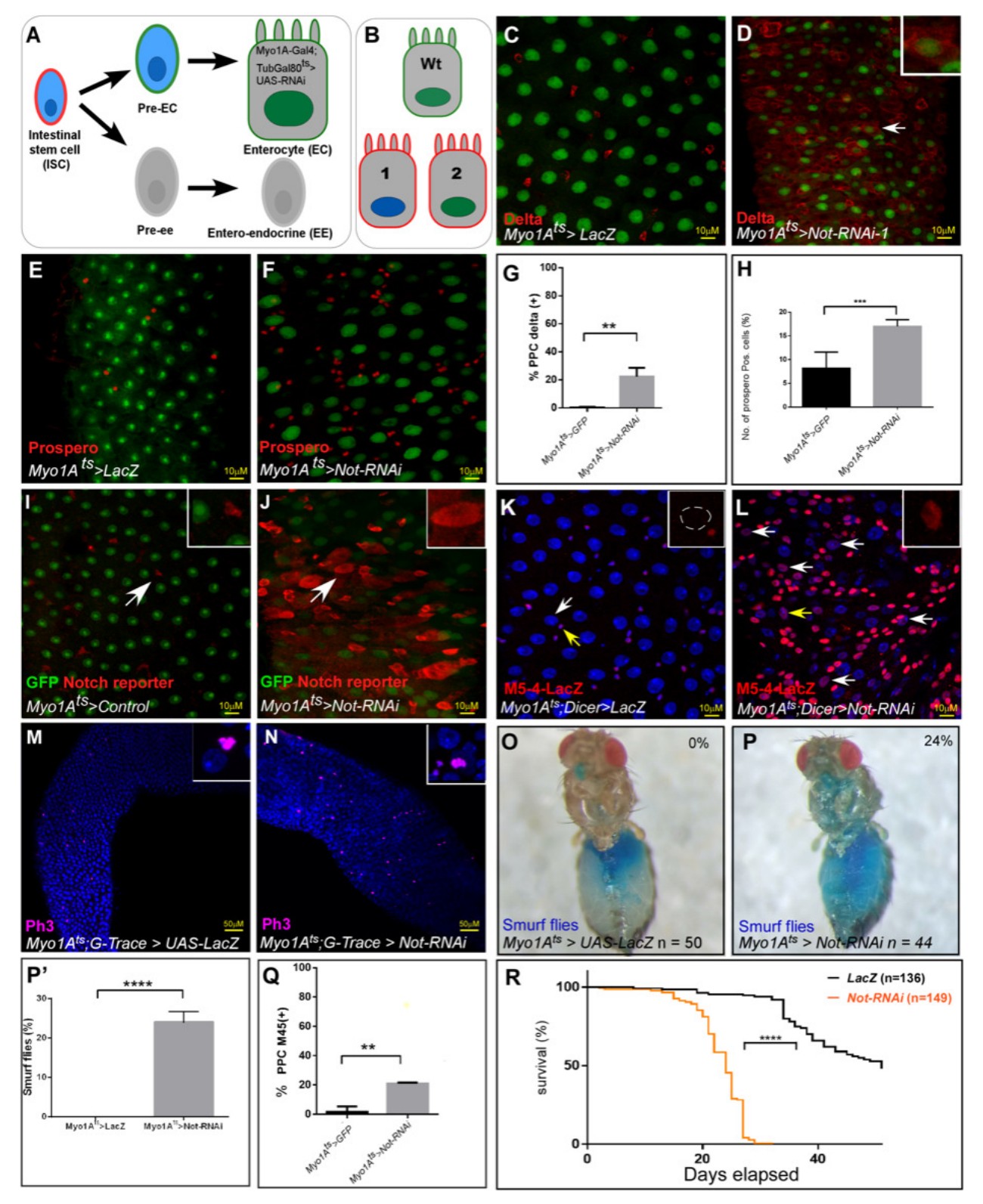

**Figure 1.** RNAi Screen identified Non-stop (Not) as an ECs identity supervisor. (**A**) Schematic diagram of midgut differentiation and an outline of the Ub/UbL screen (see text for details). The Notch ligand, Delta, is expressed on the surface of Intestinal stem cells (ISC) marked in red. (**B**) Phenotypes expected from positive hits: 1. Loss of expression of EC-specific GFP (expressed only in fully differentiated ECs using MyoIA>Gal4/Gal80ts system), along with ectopic expression of the ISC marker Delta (red). 2. Polyploid cells that ectopically express Delta and retain expression of GFP. (**C–F**)

*Figure 1 continued on next page*

*Figure 1 continued*

Confocal images using UAS-LacZ (**C, E**) or UAS-Non-stop RNAi (**D, F**) along with UAS-GFP expressed under the control of MyoIA>Gal4/Gal80$^{ts}$ system. Scale bar is 10 µM. The stem cell marker Delta (**C, D**) and EE marker Prospero (**E, F**) are shown in red. (**G, H**) Quantification of three biological repeats of experiments similar to that shown in C-F. *(**I, J**) Expression of UAS-Non-stop RNAi, but not control, in ECs for 48 hr using MyoIA>Gal4/Gal80$^{ts}$ results in ectopic expression of the Notch-reporter (red) in polyploid cells. (**K, L, Q**) Expression of the *escargot* progenitor enhancer reporter M5-4-LacZ in control or Non-stop-targeted ECs (red). Yellow arrows points to cells shown in the insets. White arrows in **L** are examples of EC-like polyploid cells ectopically expressing the reporter. (**M, N**) Loss of Non-stop in ECs resulted in an increase in the mitosis marker p-H3 in small cells. (**O, P, P'**) Loss of Non-stop in ECs impairs gut integrity as evident by the leakage of blue-colored food into the abdomen (smurf assay); 24% of Non-stop-RNAi flies show loss of gut integrity versus 0% in control flies (n = 50, 44 respectively, p<0.0001) (**Q**) Quantification of M5-4 positive PPCs in control and upon targeting Non-stop in ECs. (*** = p<0.001 **p<0.01). (**R**) Survival analysis of flies expressing the indicated transgenes in ECs under the control of MyoIA-Gal4/Gal80ts (Log-rank test ****=p<0.0001).

The online version of this article includes the following source data and figure supplement(s) for figure 1:

**Source data 1.** Summary of the screen Ub/UbL screen results.
**Source data 2.** Results of primary and secondary transgenic RNAi screens.
**Figure supplement 1.** Examples of positive hits of the Ub/UbL screen.
**Figure supplement 2.** Characterization of Non-stop in midgut cells (**A–G**).

enteroendocrine cells (EE's), or progenitor cells, using the Prospero>Gal4/Gal80$^{ts}$, or Esg>Gal4/Gal80$^{ts}$ that activates the UAS-RNAi in these cells, respectively (*Figure 1—source data 2*).

## Non-stop supervises EC identity

Among the genes identified were E3 ubiquitin ligases, E2 enzymes, SUMO-related enzymes and ubiquitin-specific peptidases (DUB/USPs). We also identified nuclear proteins harboring PHD domains that serve as bind to methylated histones, but may confer ubiquitin ligase activity present in ubiquitin ligases (Examples are shown in *Figure 1—figure supplement 1A–H*). The primary results of the screen are detailed in (*Figure 1—source data 2* including results of secondary screen). The DUB Non-stop (Not, dUSP22, CG4166) was identified as a bona-fide EC identity supervisor. RNAi mediated knockdown of Non-stop in ECs using three independent UAS-RNAi lines resulted in the inability of ECs to maintain MyoIA>UAS-GFP signal followed by ectopic expression of the ISC marker Delta on the surface of EC-like polyploid cells in both females and males (*Figure 1C and D*, quantitated in 1G and *Figure 1—source data 1*; *Figure 1—figure supplement 1H–J*).

Non-stop was discovered as a ubiquitin protease essential for axonal guidance in the visual system (*Martin et al., 1995*). Non-stop is highly conserved from yeast to humans (Ubp8 and USP22 respectively; *Mohan et al., 2014*), and it is required for deubiquitinating monoubiquitinated histone H2B (H2Bub) and activation of gene expression (*Weake et al., 2008*; *Mohan et al., 2014*; *Morgan et al., 2016*; *Li et al., 2017*).

Immunofluorescence revealed that Non-stop is expressed in all midgut cells (*Figure 1—figure supplement 2A–D*). Non-stop is the major H2Bub deubiquitinase in *Drosophila*, therefore functional loss of Non-stop should lead to an increase in H2Bub levels (*Weake et al., 2008*; *Zhang et al., 2008*; *Morgan et al., 2016*; *Mohan et al., 2014*; *Li et al., 2017*). Indeed, ECs lacking Non-stop exhibited more H2Bub, and accordingly protein extracts derived from these midguts were characterized by an over 6-fold increase in H2Bub compared to control knockdown midguts (*Figure 1—figure supplement 2F–H*). We noticed that Non-stop knockdown in ECs resulted in an increase in Prospero positive cells (likely EEs) (*Figure 1E,F*, and quantified in 1H, *Figure 1—source data 1*). However, reduction of Non-stop in EEs did not impact EE or EC identity or number, and there was no change in Delta expression. Thus, indicating that Non-stop function in maintaining differentiated identity was confined to ECs and not EEs (*Figure 1—figure supplement 2I–L*).

While we focused on differentiated midgut cells, we observed that Non-stop has also a role in the biology of progenitor cells. UAS-RNAi-mediated loss of Non-stop in progenitors (derived by the Esg>Gal4 driver that is expressed in both ISCs and EBs), resulted in increased progenitor number (Delta positive small cells). It also resulted in mis-differentiation of progenitors as evident by polyploid cells that ectopically express Delta, as well as GFP under the control of the Esg>Gal4 driver (*Figure 1—figure supplement 2M–N*). Targeting of Non-stop only in ISCs in the gut (using Delta>-Gal4), resulted in lethality at the pupal stage with no viable adults. In contrast, elimination of Non-stop in EBs using Hey>Gal4 had no impact on the gut tissue, (*Figure 1—figure supplement 2O–P*,

and not shown). However, a detailed analysis of the function of Non-stop in ISCs is outside the scope of this study that focuses on the differentiated ECs.

At the tissue level, Non-stop elimination affected the entire midgut tissue; resulting in ectopic activation of the Notch pathway, as well as the stem-cell enhancer M5-4 *esg::LacZ* in polyploid cells, indicating that these cells were losing their differentiated state (*Figure 1I–L*, quantified in 1Q, and *Figure 1—source data 1*). Loss of Non-stop also resulted in increased phospho-Histone H3 which is indicative of mitotic activity in small cells, likely progenitors (*Figure 1M,N*). Knockdown of Non-stop in ECs reduced epithelial integrity as evidenced by leaking of blue colored food outside the gut, and reduced overall survival (*Figure 1O,P and R* respectively; *Figure 1—source data 1*).

We evaluated the identity and fate of young ECs conditionally lacking Non-stop, and aged wild-type ECs, as well as the cellular composition of the gut. Towards this end we used the lineage tracing system G-TRACE. G-TRACE is a dual-color GAL4-dependent system, that enables tracing fully differentiated non-dividing cells (*Figure 2A*; *Evans et al., 2009*; *Flint Brodsly et al., 2019*). In brief, the MyoIA-Gal4/Gal80$^{ts}$ directs the expression of the color system only to fully differentiated ECs. Expression of the Gal4 is dependent on the EC-specific MyoIA promoter that induces expression of a UAS-RFP (red). Concomitantly, this Gal4 activity induces the expression of a Flp-recombinase resulting in a recombination event that drives permanent expression of a GFP regardless of the differentiation state of the cell. Therefore, wildtype young ECs express both RFP and GFP, and are the only population of polyploid cells (PPCs), observed in control midgut tissue (*Figure 2B*). In contrast, in guts where Non-stop was knocked-down in ECs, other populations of fluorescently colored PPC's were observed. These include PPCs that express only GFP, termed PPC** (PPC$^{GFP+ RFP-}$; *Figure 2C*, and quantified in 2J; *Figure 2—source data 1*). Unlike control cells, PPC** did not express EC-related transcription factors such Odd-skipped and exhibited reduced expression of the differentiated lamin, LamC (*Figure 2D–G*). We observed a small number of PPC** that express the ISC marker Delta (<%1; *Figure 2H,I*). While this is a low percentage, we never observed the expression of Delta on the surface of control PPCs. By the nature of the G-TRACE system, we concluded that these PPC** are likely ECs that were no longer fully differentiated, failing to maintain EC identity. In accordance, PPCs that did not express EC key transcription factors such as Pdm1 and caudal were also observed (*Figure 2—figure supplement 1A–D*). In addition, guts where Non-stop was targeted in ECs were populated with PPCs lacking expression of either RFP or GFP (termed PPC*), and are likely mis-differentiated progenitors that failed to activate the *myo*-promoter and the entire RFP/GFP marking system, and 30% of these cells ectopically expressed Delta (*Figure 2H–J*, *Figure 2—source data 1*).

The phenotypes observed upon acute loss of Non-stop are highly similar to the ones observed in aged midguts (*Figure 2K–R*; *Figure 2—figure supplements 2* and *3*). G-TRACE analysis of aged ECs established that the aged midgut (5 weeks old) are populated with ECs that are no longer differentiated (PPC**), as well as mis-differentiated progenitors (PPC*). These PPC** no-longer express the differentiated LamC, or the transcription factors Pdm1 and Odd-skipped and ectopically expressed the stem cell marker Delta (*Figure 2L–Q*, and quantified in *Figure 2R*; *Figure 2—source data 1*; and *Figure 2—figure supplement 3A–D*).

As in the case of young ECs lacking Non-stop, aging ECs ectopically express the stem cell enhancer M5-4 (*Figure 2—figure supplement 2A,B,I,J*). They also exhibit reduced expression of the differentiated Lamin, LamC (*Figure 2—figure supplement 2C,D,K,L*) and ectopically express the stem cell-related Lamin, LamDm0 as well as its binding partner Otefin (Ote) in PPC (*Figure 2—figure supplement 2E,F,M,N,G,H*). At the tissue level, aged ECs also exhibited disorganized distribution of EC-related adhesion molecule Discs large, (Dlg), and reduced expression of MESH and snakeskin (SSK). They ectopically express Armadillo (*Drosophila* β-catenin), which expressed on the surface of progenitors in young midguts, all resulting in loss of gut integrity (*Figure 2—figure supplement 3E–L*; Figures 9O, P). Thus, acute loss of Non-stop in young EC or age-related decline in aged ECs resulted in EC cells that lose differentiation state and/or mis-differentiated progenitors.

## A Non-stop identity complex (NIC) supervises EC identity

Non-stop is the catalytic subunit of a DUB module containing Sgf11, E(y)two and in some cases Ataxin7 that is part of the SAGA chromatin modifying complex and interacts with the DUB module (*Morgan and Wolberger, 2017*; *Lee et al., 2007*; *Mohan et al., 2014*; *Weake et al., 2008*). We therefore tested whether the SAGA complex subunits are required for maintaining EC identity. EC-

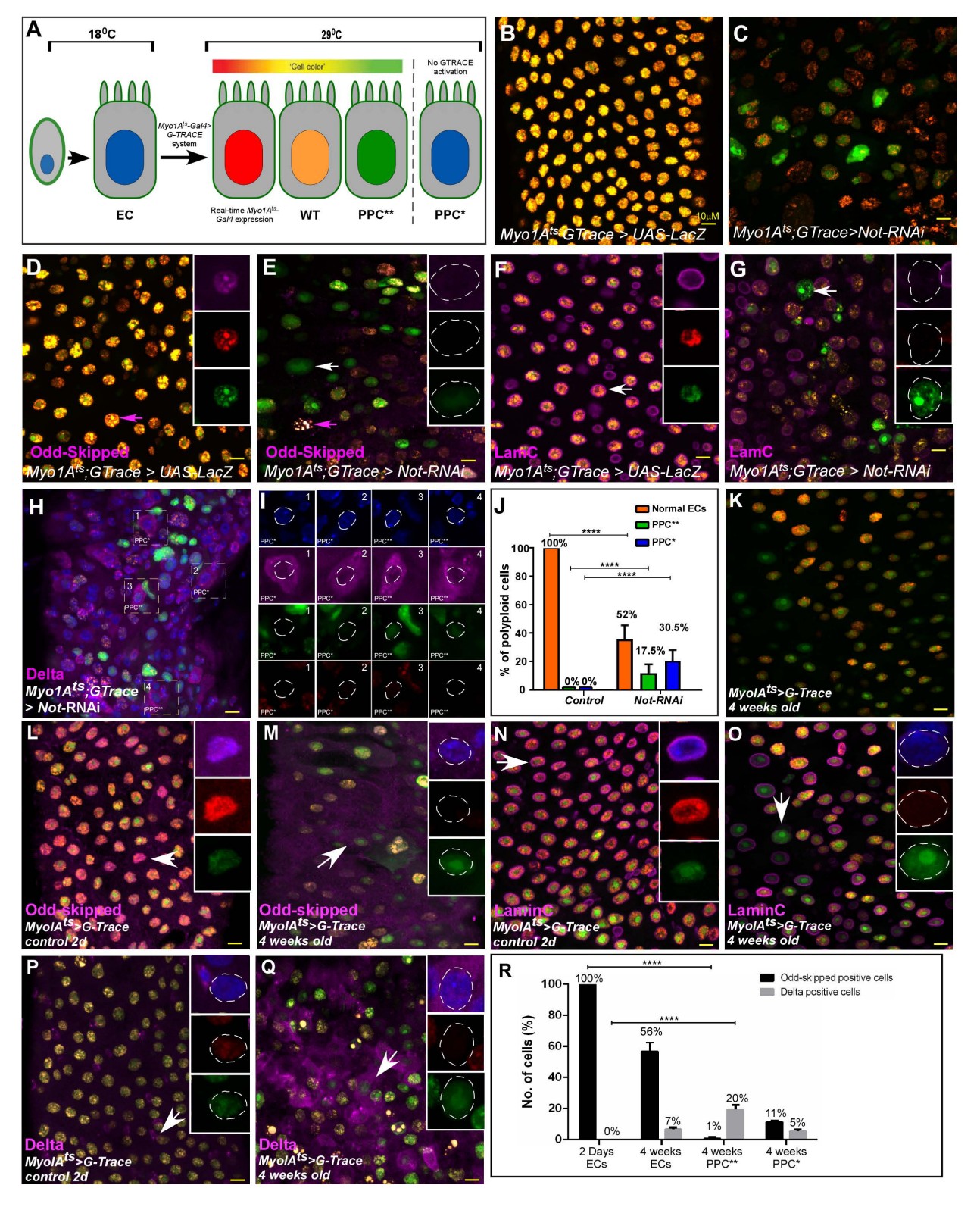

**Figure 2.** G-TRACE-Lineage characterization of Non-stop targeted young ECs and, aged ECs. (**A**) Schematic diagram of EC-G-TRACE-lineage tracing system adapted from *Flint Brodsly et al., 2019*. PPC** (RFP⁻GFP⁺), are EC that are no-longer differentiated. PPC* (RFP⁻GFP⁻) are miss-differentiated progenitors. (**B–Q**) Confocal microscopy of midguts expressing the indicated transgenes, under the control of MyoIAᵗˢ G-TRACE system using the indicated antibodies. DAPI (blue) marks DNA. Arrows point to cells shown in the insets with individual far-RFP, RFP and GFP channels. scale bar is 10

*Figure 2 continued on next page*

*Figure 2 continued*

μM. (B–G) G-TRACE of EC in control young midgut expressing either UAS-LacZ (B, D, F), or UAS-Non-stop- RNAi (C, E, G). Arrows point to cells shown in the insets with individual far-RFP, RFP and GFP channels. (H–I) Expression of Delta (purple) on the surface of the indicated PPCs. Numbered cells are shown in (I) with individual channels dashed circle outlines the nucleus. (J) G-TRACE-based quantification of PPC types (wildtype, PPC* PPC**) observed in control midguts or where Non-stop was targeted ECs. (K–Q) Confocal microscopy of midguts expressing MyoI$A^{ts}$> G-TRACE system using the indicated antibodies. (B, L, N, P) G-TRACE of EC in young, and (K, M, O, Q) old midguts. (R) Quantification of indicated PPCs expressing Odd-Skipped, and Delta similar to experiments shown in L-O (**** = p<0.0001).

The online version of this article includes the following source data and figure supplement(s) for figure 2:

**Source data 1.** Quantification data for *Figure 2J and R,*.
**Figure supplement 1.** Non-stop is required for expression of EC transcription factors, and epistatic studies with Hey (A–H).
**Figure supplement 2.** Miss-regulation of enhancers activity and nuclear Lamins in aged enterocytes.
**Figure supplement 3.** Hallmarks of aging in the *Drosophila* midgut (A–L).

specific RNAi-mediated reduction of Ataxin7 (part of the DUB module), or GCN5 (the histone acetyl transferase of the SAGA complex) did not result in loss of EC identity (*Figure 3—figure supplement 1* and not shown). We concluded that the EC identity-related function(s) of Non-stop are independent of SAGA.

Therefore, we biochemically searched for Non-stop-associated proteins that potentially together maintain EC identity. Toward this end, we generated a *Drosophila* S2 cell line stably expressing epitope-tagged Non-stop-2xFLAG-2xHA (Non-stop-FH) under the control of a copper-sulfate-responsive metallothionein promoter. Protein complexes that contained Non-stop were affinity purified using sequential capture of the epitope tags, FLAG, then HA. These complexes were subsequently resolved according to size, using gel filtration chromatography (*Figure 3A–C*, *Figure 3—source data 1*). We used a ubiquitin-AMC deubiquitinase activity assay to track enzymatically active Non-stop in the purified fractions (*Figure 3B*). We found three major peaks of deubiquitinase activity. The major activity peak resolved at about 1.8 MDa, together with components of SAGA complex (Group 1). A second peak was resolved centering approximately around 670 kDa (Group 2). A third peak, with the lowest total activity, was detected centering around 75 kDa. The three fractions comprising the center of each peak were combined and constituent proteins identified by mass spectrometry (MudPIT) (*Washburn et al., 2001*). Group two contained Non-stop, e(y)2, and Sgf11 (the DUB module), but no other SAGA subunits. It also contained members of a known boundary complex that includes Cp190, Nup98, Mod (Mdg4) and is known to be part of nuclear complex regulating enhancer-promotor interactions and affecting transcriptional memory (*Pascual-Garcia et al., 2017*). In both groups 1 and 2 Histones H2A and H2B were also detected, showing that the DUB module was co-purifying with known substrates of Non-stop and indicating the DUB was purifying in a physiologically native state (*Figure 3C*; Zhao et al., 2008).

We further studied the interaction between Non-stop and members of the protein complex that we termed NIC (Non-stop identity complex), using endogenous co-immunoprecipitation from fly-derived protein extracts, in vitro binding assay, and yeast direct-hybrid systems (*Figure 3D–H*, *Figure 3—figure supplement 1F*). As shown in *Figure 3D* co-immunoprecipitation experiments established interaction between endogenous Non-stop and Cp190 as well as with Mdg4. In vitro binding, using S2 cell-derived extract expressing HA-Non-stop and His-tagged bacterially expressed and purified proteins, confirmed that Non-stop interacted with its known interaction partner E(y)2. Non-stop also interacted with Cp190 via the C-terminal portion of Cp190 [amino acids (a. a.) 468–1096], and only minimally with the N-terminal portion of Cp190 (a.a. 1–524) (*Figure 3E,F*). Using Y2H we mapped the interaction between Cp190 and Non-stop to the second and third zinc fingers of Cp190 but not the first or fourth (*Figure 3G*). Non-stop did not interact with Mdg4 in in vitro binding assay or in a simple Y2H. Suggesting that the interaction between Non-stop with Mdg4 likely involves the entire DUB module (e.g. Sgf11 and E(y)2). Indeed, once all three proteins are co-expressed in a yeast cell (yeast three hybrid system, (Y3H), an interaction with Mdg4 is detected (*Figure 3—figure supplement 1F*)). We modified the resulting Y3H and observed that in yeast cells it is possible to show the assembly a complex consisting all four proteins (Y4H; *Figure 3—figure supplement 1G*). Using this system, we determined that the interaction between Mdg4 with the DUB ternary complex (Eny2/Sgf11/Non-stop) requires a region adjacent to the BTB domain of Mdg4 (*Figure 3G*). While we were not technically able to test the binding of Nup98 to other members of the complex it was

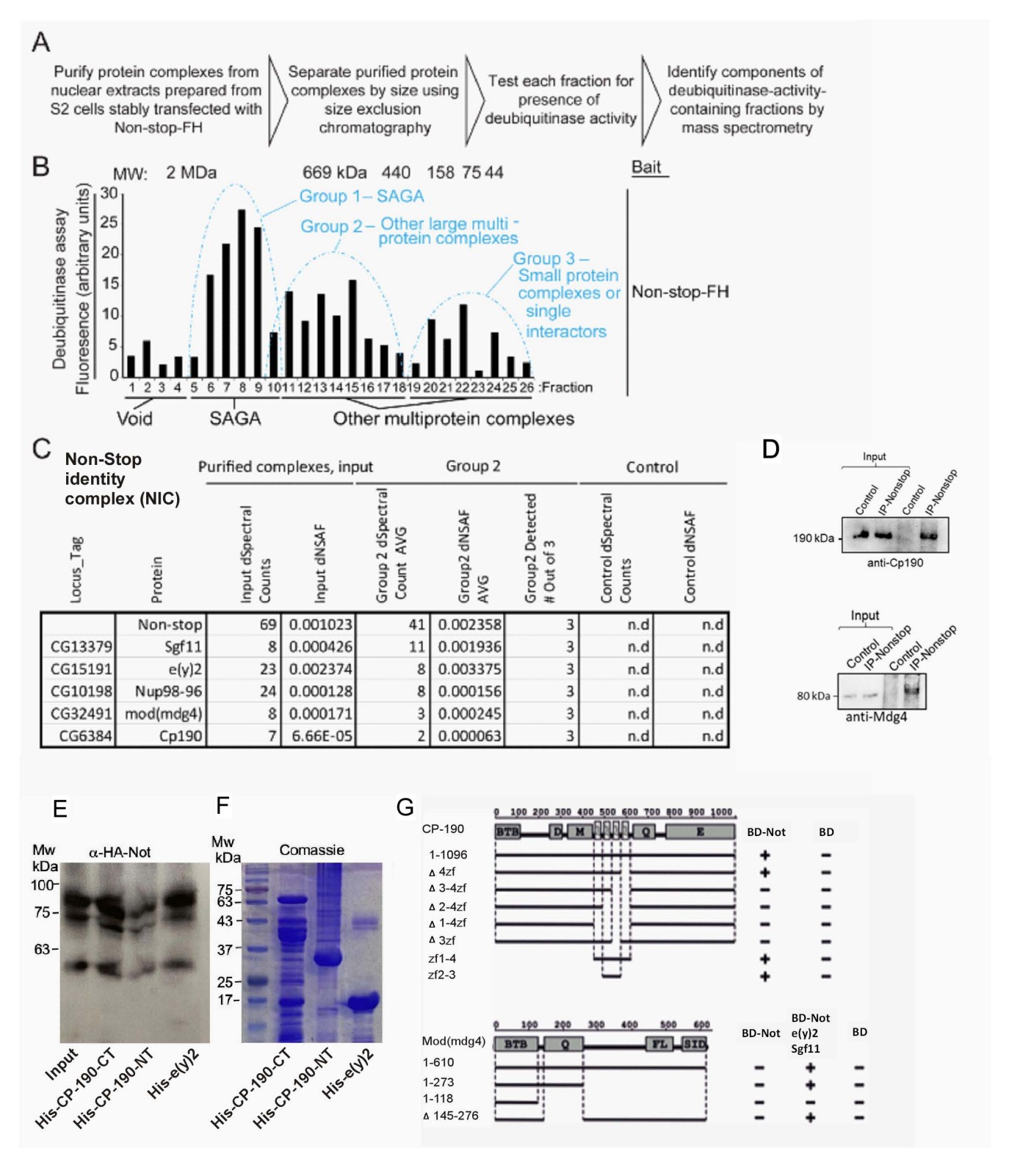

**Figure 3.** Identification of a Non-stop-identity complex (NIC). (**A–C**) Purification scheme of nuclear Not-associated complexes from *Drosophila* S2 cells. (see text and methods; Reproduced from *Figure 1B and E*, *Cloud et al., 2019*), eLife, published under the Creative Commons Attribution 4.0 International Public License (CC BY 4.0; https://creativecommons.org/licenses/by/4.0/). (**B**) Identification of Not-associated isopeptidase activity proteins by immunoprecipitation followed by size fractionation and mass-spectrometry. CP190, Mod (mdg4), Nup96-98, and E(y)two were all present in Group 2. *Figure 3 continued on next page*

*Figure 3 continued*

Not-FH; IP with full length Not FLAG-HA tagged (**C**) Summary of protein complexes isolated identified by mass-spectrometry (**D, E**) Not binds to the C-terminal portion of CP190 and to E(y)2. (**D**) Immunoprecipitation confirms Non-stop specifically interacts with NIC subunits Cp190 and Mdg4. Endogenous Non-stop was immunoprecipitated from whole cell extracts prepared from adult OregonR flies using an α-Non-stop antibody. Control immunoprecipitations were performed with α-guinea-pig IGG. The presence of NIC subunits was assayed by immunoblotting with antibodies specific for Cp190 or Mdg4 as indicated. (**E**) Western-blot of in vitro binding between HA-Not derived from S2 cell extract and the indicated bacterially expressed purified His-tagged proteins. 10% input is shown. (**F**) Coomassie blue staining of the indicated bacterially expressed His-tagged proteins used in the binding assay in (**E**). (**G**) Upper panel: Schematic diagram of Y2H interaction assay between CP190 and Non-stop. Different fragments of CP190 were fused to the activation domain (AD) of GAL4 and tested for interaction with Non-stop fused to the DNA-binding domain (BD) of GAL4. Protein domains of full-length CP190 are indicated as boxes, and lines represent the different deletion fragments. Zf denote zinc-fingers; BTB, BTB/POZ domain; D, aspartic acid -rich region; M, microtubule-interacting region; E, acid glutamate-rich region of CP190. The results are summarized in columns on the right (BD-Not and BD alone), with the '+" and "- "signs denotes presence and absence of interaction, respectively. Lower panel: Schematic diagram of Y2H and Y4H interaction assay between Mod(mdg4) and Non-stop. Different fragments of Mod(mdg4) were fused to the activation domain (AD) of GAL4 and tested for interaction with Non-stop fused to the DNA-binding domain (BD) of GAL4 and complex of BD-Non-stop with Eny2 and Sgf11. Protein domains of full-length Mod(mdg4) are indicated as boxes, and lines represent the different deletion fragments. BTB, BTB/POZ domain; Q, glutamine-rich region; FL, FLYWCH-type zinc finger domain; SID, Su(Hw) interaction domain. The results are summarized in columns on the right (BD-Not, BD-Not/Eny2/Sgf11 and BD alone), '+" and "- "signs denotes presence and absence of interaction, respectively.

The online version of this article includes the following source data and figure supplement(s) for figure 3:

**Source data 1.** Proteomic analysis of Non-stop bound proteins.
**Figure supplement 1.** SAGA subunits and Su(Hw) do not regulate EC identity.

previously shown that Nup98 interacts with Cp190 (*Pascual-Garcia et al., 2017*). However, further analysis will be required to fully establish the exact interactions interphases of proteins within the NIC complex.

We hypothesized that if NIC supervises EC identity, RNAi-mediated elimination of each of its subunits will result in loss of EC identity similar to the loss of Non-stop. Indeed, EC-specific knockdown of all NIC subunits except Sgf11 resulted in loss of identity and inability to maintain expression of the EC gene LamC (*Figure 4A–F*, *Figure 3—figure supplement 1*). It also resulted in ectopic expression of Delta (*Figure 4G–L*). In contrast, loss of Su(Hw), an insulator protein that binds to Mdg4 but was not identified as a Non-stop binding partner, did not result in any detectable phenotype (*Figure 3—figure supplement 1E*).

## Non-stop supervises EC-gene signature and regulates chromatin accessibility

Non-stop is well known to regulate gene expression (*Mohan et al., 2014*; *Li et al., 2017*). To elucidate Non-stop-dependent expression signatures, we determined the changes in transcriptional expression using RNA-Seq and its effect(s) on chromatin accessibility by ATAC-seq analyses (*Figure 5*, *Figure 5—source datas 1* and *2*). We determined the changes in transcription signatures of whole guts upon elimination of Not in ECs using UAS-Not-RNAi and the EC-specific MyoIA-Gal4[ts]. We identified 863 genes exhibiting downregulated mRNA expression upon loss of Non-stop in ECs (*Figure 5*; *Figure 5—source data 1*). Of these, 38% (326/863) were previously identified as EC-related genes (*Figure 5A*; *Korzelius et al., 2014*). Metascape analysis unveiled that these shared targets consist of core EC pathways that execute many of the physiological tasks of the gut (*Figure 5B*; *Figure 5—source data 1*). In addition, 51% (444/863) of Nonstop down regulated genes were not previously identified as EC genes in other studies. However, metascape analysis showed this group of genes is also enriched for EC-related pathways and physiology (*Figure 5—figure supplement 1A*). Among Non-stop down-regulated genes were transcription factors including Odd-Skipped, and Relish, which is the central transcription factor of the IMD-innate immunity pathway that is involved in the local innate response of the gut to infection (*Figure 5—figure supplement 1A2, A3*).

We previously identified genes that require Hey for their expression in ECs. Here, we found that 76% (174/228) of Hey-dependent genes also required Non-stop for expression (*Figure 5C*). However, our expression data suggests that neither Non-stop nor Hey regulate the expression of one another (*Figure 5—source data 1*; *Flint Brodsly et al., 2019*). Performing epistatic experiments, we found that targeting Non-stop along with expression of Hey in ECs did not suppress Non-stop loss

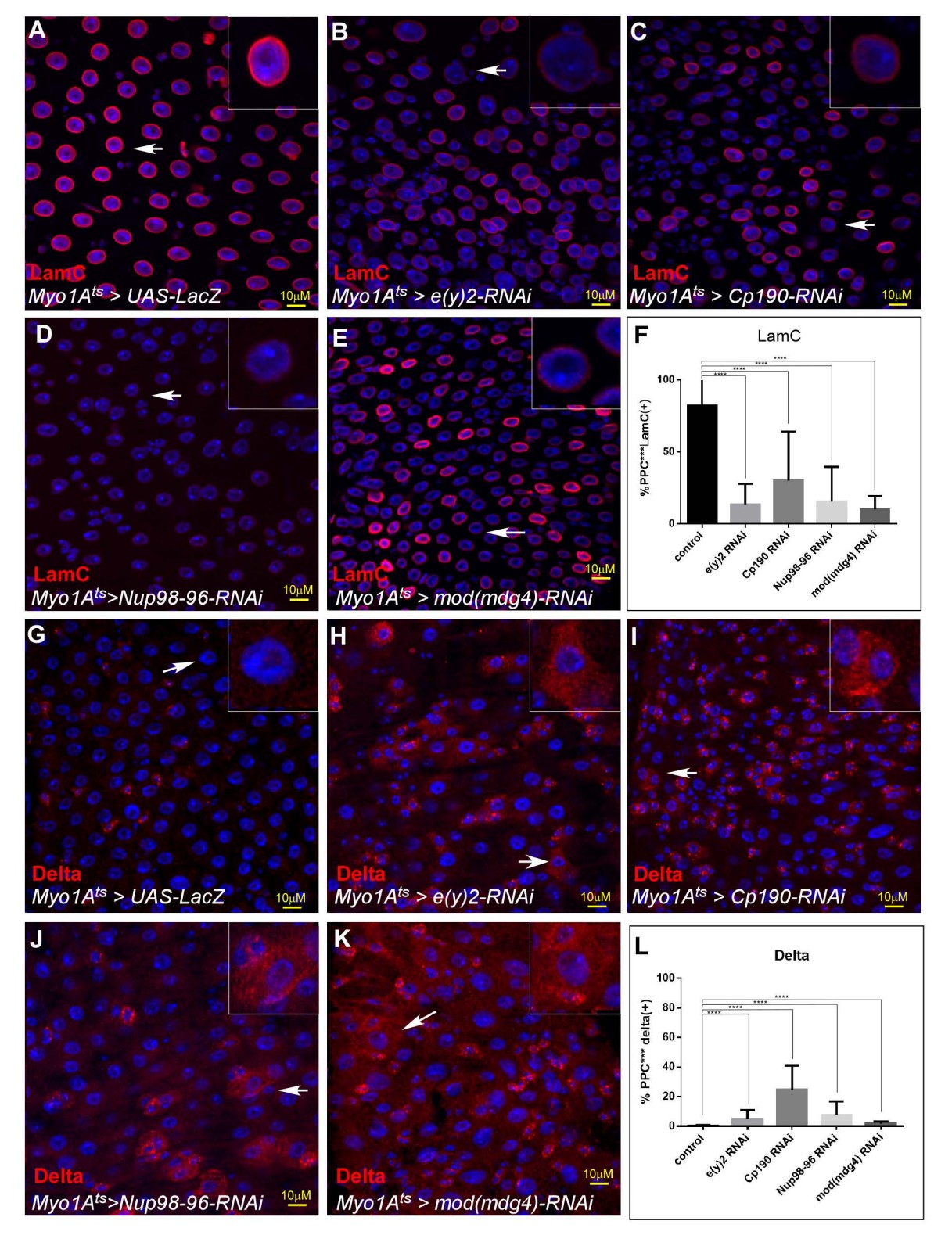

**Figure 4.** Non-stop identity complex (NIC) regulates EC identity. Confocal images of the midgut tissue using the indicated antibodies; (A–E) LamC, (G–K) Delta, DAPI marks DNA (blue). The indicated transgenes were expressed in EC using the MyoIA-Gal4/Gal80ts system for forty-eight hours. (A, G) UAS-LacZ (B, H) UAS-e(y)2-RNAi; (C, I) UAS-CP190-RNAi;. (D, J) Nup98-96 (E, K) Mod (mdg4) White arrows points to cells shown in insets, and scale bar is 10 μM. Quantification is shown in (F) for LamC. and (L) for Delta.

*Figure 4 continued on next page*

*Figure 4 continued*

The online version of this article includes the following source data for figure 4:

**Source data 1.** Quantification of cell populations described in 4F, 4L.

of identity phenotypes. Likewise, expression of HA-Non-stop along elimination of Hey in ECs did not restore EC identity (*Figure 2—figure supplement 1E–H*).

The expression of EC-specific genes was repressed by ectopic expression of the ISC-related lamin, LamDm0, in ECs (*Flint Brodsly et al., 2019*). Indeed, fifty percent (188/372) of genes that are repressed by expression of LamDm0 in ECs also required Non-stop for their expression, and 46 of these genes were regulated also with Hey (*Figure 5D*, *Figure 5—figure supplement 1D* and see discussion). However, the exact nature of interaction between Non-stop and Hey, and the potential co-regulation of EC-shared targets requires additional studies.

In parallel, we examined whether expression of EC-genes involves Non-stop-dependent regulation of chromatin accessibility using ATAC-seq (*Figure 5—source data 2*). We identified 214 loci that exhibited reduced chromatin accessibility ('closed'). Of these, 75% (162/214) were located in the range of 0–10 Kb vicinity of genes that exhibited reduced expression (*Figure 5A*; *Figure 5—figure supplement 2A*, and *Figure 5—source data 2*). STRING gene ontology analysis of these 'closed' regions suggested that they belong to genes that maintain the physiological properties of enterocytes (*Figure 5E*). Alignment of the 'closed' chromatin regions showed that they cluster to discrete gene regions (*Figure 5F*; for EC-related down-regulated genes, and *Figure 5—figure supplement 2A* for all closed sites; *Figure 5—figure supplement 2—source data 1*). As shown in *Figure 5F*, one cluster was located to the 5' UTR, a second cluster was at the transcriptional start site (TSS), a third was spanning the coding region, and a fourth was located at the 3'-UTR. MEME analysis revealed that they are statistically significantly enriched in DNA motifs that are known binding sequences of TFs (*Figure 5F*, *Figure 5—figure supplement 2B*).

In addition, 565 genes showed upregulation of mRNA expression upon loss of Non-stop, and are related to progenitor fate, cell cycle, and DNA repair (*Figure 5—figure supplement 1D*). In a sharp contrast to the numerous closed regions only a small number (~16) regions exhibited increased accessibility upon loss of Non-stop in ECs interestingly many of these genes code for long non-coding RNA (*Figure 5—figure supplement 2C*). Thus, supporting the notion that Non-stop acts primarily to maintain chromatin accessibility in the vicinity of its targets.

We hypothesized that the ectopic expression of these genes may be, at least partially, due to changes in nuclear organization in ECs. In this regard, among the genes that require Non-stop/NIC at the protein level is LamC (*Figure 4A–F*). LamC is the dominant lamin in ECs that silences the expression of stem cell and non-relevant gene programs in ECs (*Flint Brodsly et al., 2019*). For example, PCNA is not expressed in control ECs, but is ectopically expressed in PPC** (ECs that are no longer differentiated; PPC$^{GFP+ RFP-}$; *Figure 5—figure supplement 1F,G*) when non-stop is silenced. This ectopic expression was prevented by co-expression of LamC in ECs where Non-stop was eliminated (*Figure 5—figure supplement 1H*). Moreover, loss of Non-stop in ECs also resulted in a significant decrease in the linker histone H1 that is associated with compacting chromatin and gene silencing (*Fyodorov et al., 2018*). H1 protein levels were reduced in the nuclear periphery of ECs lacking Non-stop (*Figure 5G,H*) in gut extracts derived from flies where Non-stop was targeted in ECs, as well as upon knockdown of Non-stop in S2R cells (*Figure 5I,J* respectively). Thus, the ectopic expression of non-EC programs may be due to loss of LamC and H1 proteins and subsequently heterochromatin impairment and dependent silencing. However, since we isolated mRNA from the entire midgut, the source of these upregulated mRNAs may also be from mis-differentiated stem cells (PPC*), as well as from the increase in rapidly dividing progenitor cells.

Comparison of Non-stop RNA-seq data with a genome-wide high-resolution DamID binding map of histone H1 performed in Kc167 *Drosophila* cells (*Braunschweig et al., 2009*) identified the GAGAGA sequence that is the binding sites for the transcription factors Trithorax-related (Trl/GAF), a shared motif for Non-stop-regulated genes also bound by H1. Moreover, GAGAGA sequence was also enriched in Non-stop closed regions at the TSS of genes requiring Non-stop for expression (*Figure 5F*).

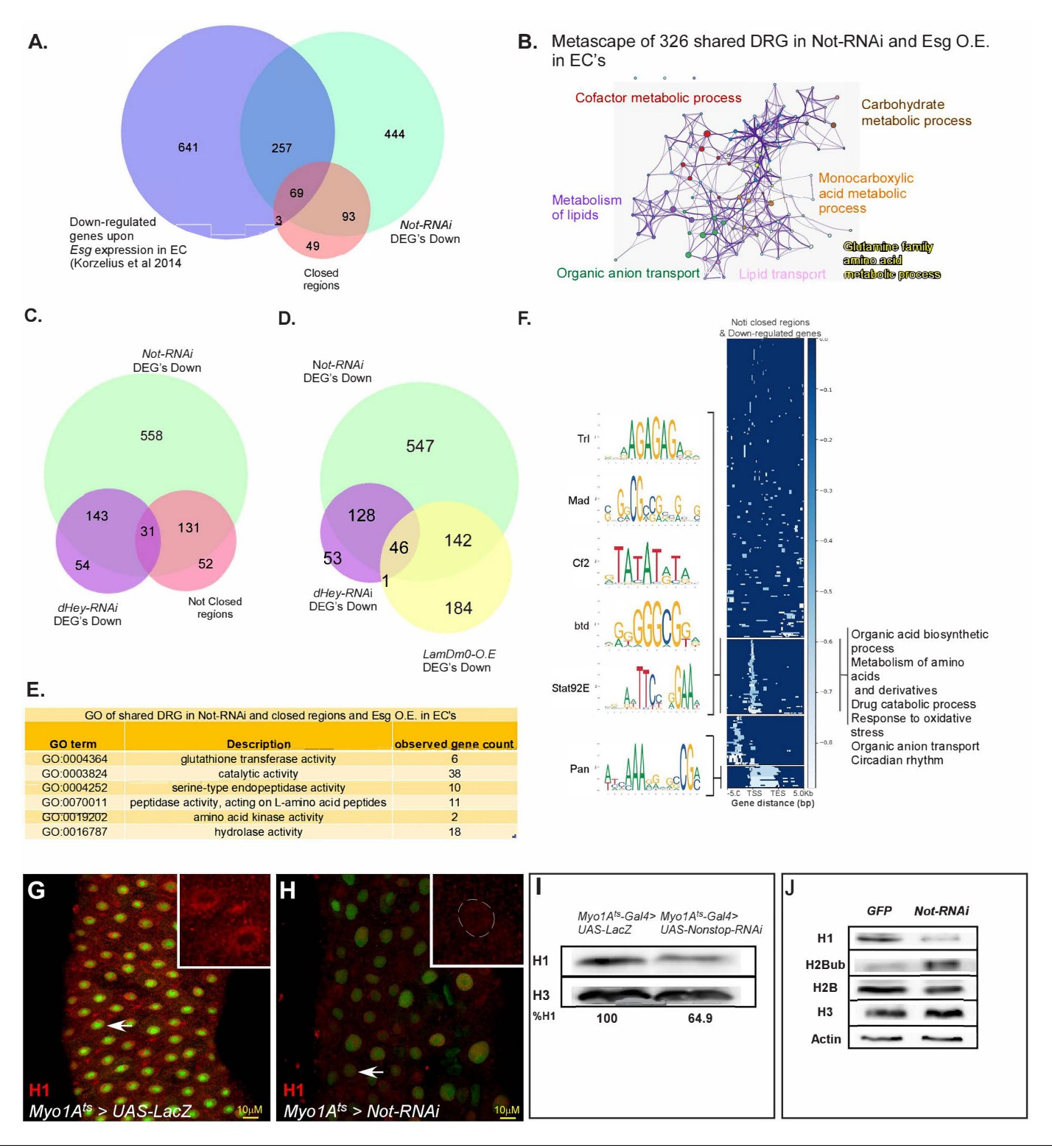

**Figure 5.** Not regulates EC-gene expression and is required for chromatin accessibility. (**A**) Venn diagram comparing EC-related genes (Blue; *Korzelius et al., 2014*), genes exhibiting reduced expression upon loss of Not in ECs (Green), and chromatin regions with reduced accessibility upon loss of Not in ECs identified by ATAC-seq (Orange). (**B**) Metascape analysis of shared Non-stop-down-regulated genes and Esg over-expression in ECs. (**C**) Venn diagram comparison of genes that exhibit reduced expression upon loss of either Non-stop or Hey in ECs, as well as genes in the vicinity of regions showing reduced accessibility upon loss of Non-stop. (**D**) Venn diagram of genes that exhibit reduced expression upon loss of Non-stop or Hey and of genes with reduced expression upon over expression of LamDm0 in ECs (**E**) GO analysis of genes downregulated by loss of Non-stop in and

*Figure 5 continued on next page*

*Figure 5 continued*

Esg over-expression in ECs exhibiting reduced accessibility. Observed gene count; number of genes identified from this group in both ATAC-seq and RNA-seq (F) Genome-wide alignment and MEME analysis of regions with reduced accessibility in the vicinity of down-regulated genes upon loss of Non-stop in ECs. TSS, transcriptional start site; TES, transcription end site. (G–H) Confocal images of the midgut tissue using α-Histone H1 (red), and expressing the indicated transgenes in ECs using the MyoIA-Gal4/Gal80ts system for forty-eight hours, DAPI marks DNA (blue). (G) UAS-LacZ (control) (H), UAS-Non-stop RNAi. Scale bar is 10 μM. (I, J) western-blot analysis of the indicated proteins derived from gut extract (I), or S2 *Drosophila* cell extract (J) Histone H3 and Actin serve as loading controls.

The online version of this article includes the following source data and figure supplement(s) for figure 5:

**Source data 1.** RNA-seq of Non-stop-regulated genes.
**Source data 2.** ATAC-seq profiling of non-stop dependent changes in chromatin accessibility.
**Figure supplement 1.** Analysis of Not-related RNA-seq and ATAC-seq.
**Figure supplement 2.** Analysis of changes in chromatin actability upon loss of Not.
**Figure supplement 2—source data 1.** Gene clustering of Non-stop closed regions (complement *Figure 5—figure supplement 2*).

While Trl/GAF was not identified as part of the NIC in our proteomic purification, it associates in a protein complex containing Nup98, e(y)two and Mod (mdg4) that regulate gene-expression (*Pascual-Garcia et al., 2017*). However, targeting Trl/GAF or the adaptor protein CLAMP protein which also binds to the GAGAGA sequence and associates with Cp190 and Mod (mdg4) did not result in loss of EC identity (Not shown; *Bag et al., 2019*). Therefore, we suggest that NIC is likely functionally and compositionally distinct from the Trl-containing complex.

## Non-stop maintains the protein level of NIC subunits

Non-stop/USP22 iso-peptidase activity is required for transcriptional activation, as well as for preventing ubiquitin-dependent degradation (*Atanassov et al., 2009*; *Atanassov and Dent, 2011*; *Cloud et al., 2019*). Thus, it seemed possible that Non-stop is required for either the mRNA expression of NIC subunits, or alternatively for protecting NIC subunits from proteasomal degradation. As evident by our RNA-seq analysis, the mRNA level of NIC subunits was not affected by loss of Non-stop in EC (*Figure 5—source data 1*). However, the protein levels of Cp190, e(y)2, Mod (mdg4), and Nup98 were reduced upon RNAi-dependent Non-stop elimination in young ECs as observed by immunostaining (*Figure 6A–L*; *Figure 6—source data 1*). Moreover, Nup98 was no-longer confined to the nuclear envelope but was localized to the nucleus interior in a punctate pattern (*Figure 6E,F*; *Figure 6—source data 1*). Taken together these results suggest that the regulation of NIC subunits by Non-stop is at the protein level. However, a detailed biochemical study demonstrating a direct stabilizing impact on the stability of NIC subunits by Non-stop is currently being investigated.

The observed changes in the stability of LamC and NIC subunits encouraged us to examine the larger organization of the nucleus using proteins that are markers for specific intranuclear domains and bodies. Loss of Non-stop in ECs resulted in decline in Coilin, which resides within Cajal bodies, and expansion in the expression of nucleolar Nop60B, a marker of the nucleolus. At the nuclear periphery we noted changes in localization of mTor, subsequent decline in LamC and reduced protein level of HP1b that is associated with heterochromatin and the chromocenter (*Figure 6M–T*; *Figure 2F,G*). Moreover, these changes were also observed upon targeting individual NIC subunits (*Figure 6—figure supplement 1*).

## Non-stop expression in aged ECs restores large-scale nuclear organization of ECs and suppresses aging phenotypes

The observations described above suggested a physiological relevance of Non-stop failure to aging (*Rodriguez-Fernandez et al., 2020*). The cellular and tissue phenotypes associated with acute loss of Non-stop highly phenocopy aged midguts (*Figure 2—figure supplements 2* and *3*). Indeed, a decline in the protein level of Non-stop was observed in aged ECs (*Figure 7A,F,P*). Moreover, the protein levels of NIC subunits CP190, e(y)2, and Mod (mdg4) were also reduced in aged ECs (compare *Figure 7B-E-G-J*; quantified in *Figure 7Q–T*; *Figure 7—source data 1*). Thus, suggested that a decline in Non-stop protein resulting in a failure to safeguard NIC stability accompanies aging. Therefore, we tested whether preventing decline in Non-stop protein can protect from loss of NIC. Towards this aim we continuously expressed Non-stop in ECs using UAS-non-stop and the MyoIA>-Gal4/Gal80ts system, expressing Non-stop to a level similar to its expression in young ECs as

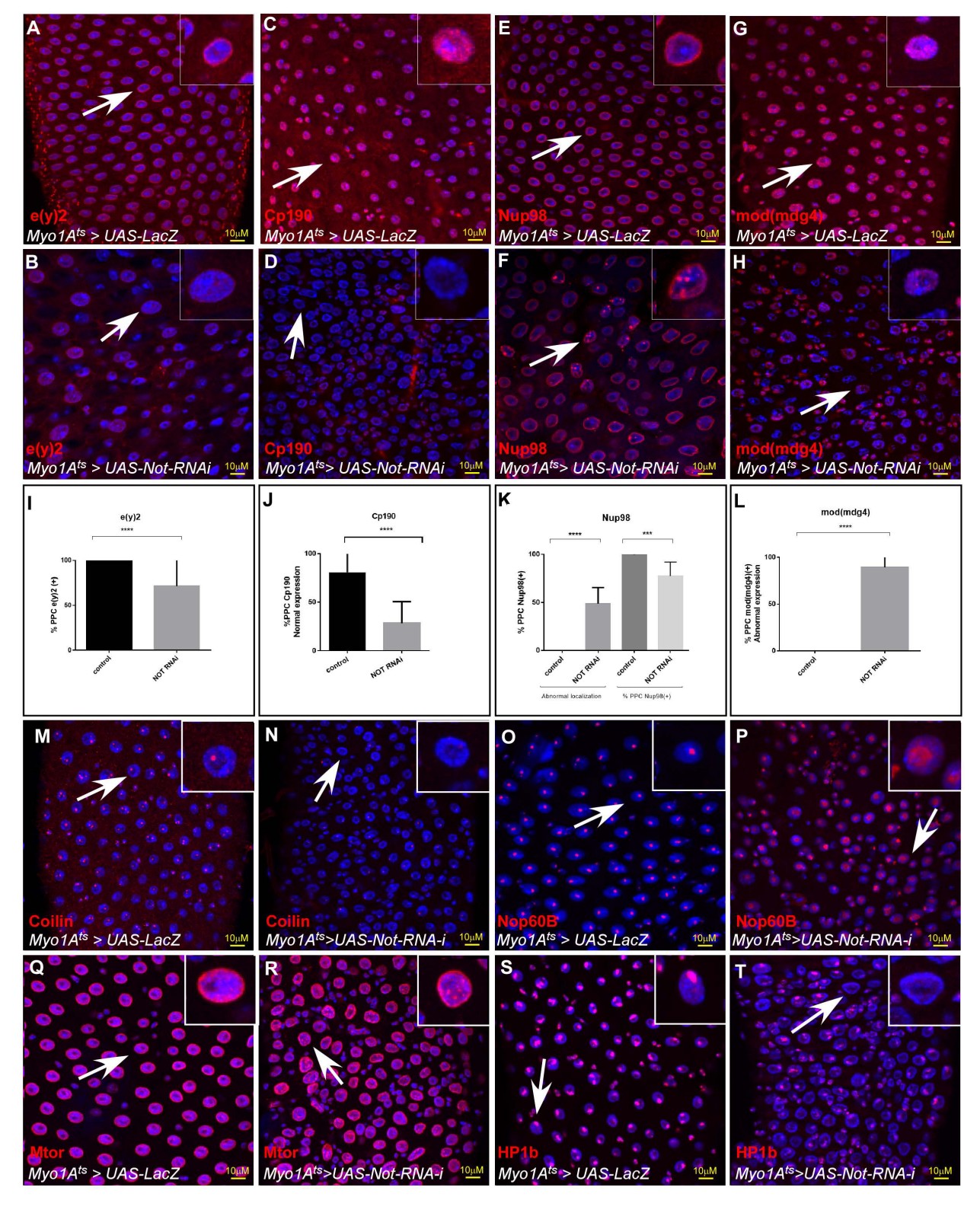

**Figure 6.** Non-stop maintains the protein level and intranuclear localization the NIC subunits, (A–H). (A–H) Representative confocal images of the midgut tissue using the indicated antibodies (red) and expressing the indicated transgenes in EC using the MyoIA-Gal4/Gal80ts system. UAS-LacZ (A, C, E, G), UAS-Non-stop RNAi (B, D, F, H). DAPI marks DNA (blue), and scale bar is 10 µM. White arrows points to cells shown in insets, and scale bar is 10 µM. (I–L) Quantification of 3 biological experiments is shown (M–T) Non-stop regulate large-scale organization of the nucleus. Representative

*Figure 6 continued on next page*

*Figure 6 continued*

confocal images of the midgut tissue using the indicated antibodies (red) and expressing the indicated transgenes in EC using the MyoIA-Gal4/Gal80ts system. UAS-LacZ (**M–P**), UAS-Not RNAi (**Q–T**).

The online version of this article includes the following source data and figure supplement(s) for figure 6:

**Source data 1.** Quantification of cell populations described in 6I-L.

**Figure supplement 1.** NIC subunits are required for maintaining large-scale organization of the EC nucleus.

---

determined by immunofluorescence (*Figure 7*). Indeed, and consistent with Non-stop's role as a key stabilizer of NIC, expression of Non-stop, but not the control (UAS-LacZ), for five weeks prevented the aged dependent-decline of individual NIC subunits (*Figure 7K–O*; quantified in *Figure 7P–T*; *Figure 7—source data 1*).

We further examined whether maintaining Non-stop protein levels attenuates aging of the midgut using the above system. Aging is associated with distorted nuclear organization of ECs (*Figure 8*, *Figure 2*, *Figure 2—figure supplement 2*). These changes include re-organization of the nuclear periphery, including a reduced level of LamC and histone H1 as well as redistribution of mTOR (*Figure 8A–I*). Aging is also associated with ectopic expression of LamDm0, and re-localization to the nuclear periphery of LamDm0 binding partner, Otefin (Ote) (*Figure 8J–L*, and *Figure 8—figure supplement 1G–I*). Changes are also observed in the nucleus interior, involving the nucleolus and Cajal bodies as observed by the expansion of the nucleolar protein Nop60B and Coilin, which are resident proteins in these sub-nuclear bodies (*Figure 8M–O*; *Figure 8—figure supplement 1A–C*). Consistent with Non-stop as a key identity supervisor relevant to aging, continuous expression of Non-stop for five weeks suppressed the above age-related changes in nuclear organization. Non-stop expression greatly restored LamC, histone H1 levels, localization of Mtor and suppressed the ectopic expression of LamDm0 and Ote, as well as restored the large-scale organization of the aged EC nucleus (*Figure 8C,F,I,L,O* and *Figure 8—figure supplement 1B, E, H*).

We further tested whether expression of Non-stop in wild-type ECs is capable of attenuating age-related changes in the gut epithelia. Indeed, expression of Non-stop suppressed classical characteristics of the aged midgut. For example, Non-stop expression maintained the expression of the EC marker Myo>GFP and suppressed the ectopic expression of the ISC marker Delta in PPCs (*Figure 9A–D,H*). Continuous expression of Non-stop in wildtype ECs for five weeks restored the expression of EC-related junctional proteins SSK and MESH, and suppressed the ectopic expression of Arm in PPCs (*Figure 9E–G and I–N*). To test whether Non-stop maintenance affected the entire midgut at the organ level we tested for overall gut integrity using the Smurf assay. We observed that continuous expression of Non-stop in ECs greatly prevented the extensive leakage of blue-colored food observed in five weeks old aged animals restoring gut integrity (*Figure 9O–R*).

Thus, Non-stop is required for expression of EC-gene programs, stabilizes NIC subunits in the adult, and together with NIC regulates large-scale organization of the differentiated nucleus, safeguarding EC identity and protecting from premature aging.

## Discussion

### An in vivo screen identifies regulators that maintain the differentiated state

We performed an identity screen focused on conserved enzymes within the ubiquitin and ubiquitin-like pathways using the midgut tissue as a model system. Based on cell-specific secondary tests, we identified three categories of supervisors; 1. EC-specific identity regulators 2. Genes that are required for differentiated cell identity (both EEs and ECs) 3. Genes that are required for identity of all cell types (general identity regulators). Of specific interest were a group of genes (CG1490; CG2926; CG4080) whose elimination in EEs resulted in a loss of EC, but not EE, identity, acting as inductive identity regulators. Likely their effect on ECs involves cell~cell communication via diffusible factors. While loss of each individual gene identified in the screen resulted in loss of EC identity, the molecular and genetic connections between different identity regulators requires further studies.

The observation that Ub/UbL-related genes protect the differentiated identity is conserved across species. Screens in mammalian systems identified enzymes within the SUMO and ubiquitin pathways

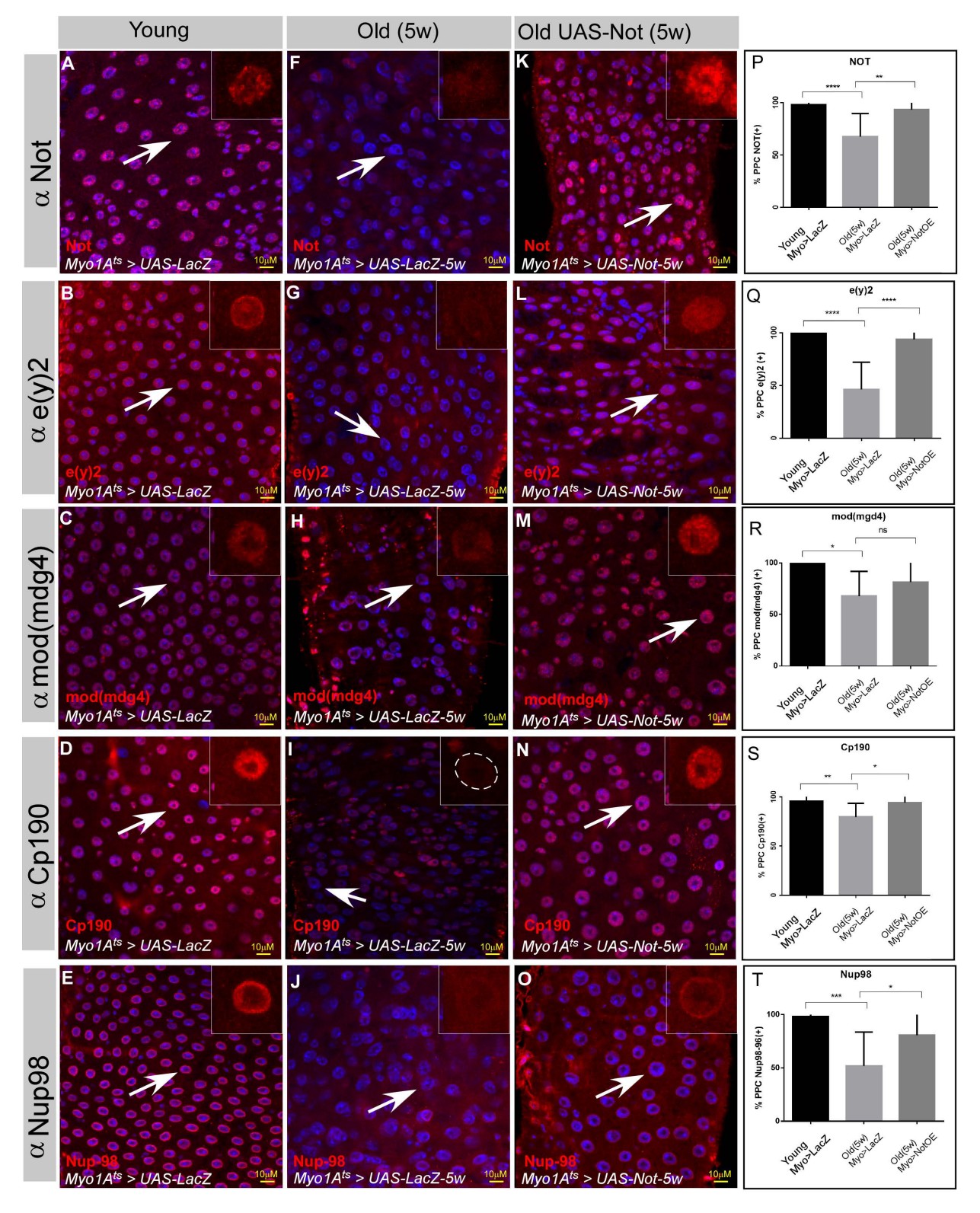

**Figure 7.** The protein levels of the Not-CP190 complex subunits decline upon aging and is restored upon continues expression of Non-stop in aged ECs. (A–O) Representative confocal images of the midgut tissue using the indicated antibodies (red) and expressing the indicated transgenes in EC using the MyoIA-Gal4/Gal80ts system. (A–E) Young 4 days old guts, (F–J) Five weeks old guts expressing UAS-lacZ. (K–O) Five weeks old guts
*Figure 7 continued on next page*

*Figure 7 continued*

expressing UAS-Non-stop. DAPI marks DNA (blue), and scale bar is 10 µM. (P–T) Quantification of similar experiments presented in A-O. **** = P < 0.0001, ***P < 0.001; **P<0.01; *=P<0.1.

The online version of this article includes the following source data for figure 7:

**Source data 1.** Quantification of cell populations described in 7P-T.

acting as a barrier against forced reprogramming of differentiated cell. Among these genes were Ubc9, the sole SUMO conjugating enzyme, that was also identified in our screen and the isopeptidase Psmd14 (*Cheloufi et al., 2015*; *Buckley et al., 2012*). In addition to Non-stop, our screen identified the iso-isopeptidase UTO6-like (CG7857), Usp7, and Rpn11 as regulators of EC identity. Rpn11 is part of the lid particle of the 26S proteasome, involved in deubiquitinating proteins undergoing proteasomal degradation (*Greene et al., 2020*).

Genes regulating identity are likely to serve as a barrier to tumorigenesis having a tumor suppressive function. The human ortholog of Non-stop, USP22 has mixed oncogenic and tumor-suppressive functions (*Jeusset and McManus, 2017*). Relevant to our study is the observation that USP22 has tumor suppressive functions in colon cancer by reducing mTor activity (*Kosinsky et al., 2020*). Along this line it is interesting to note that 10/17 of human orthologs to genes discovered in our screen are either mutated or silenced in cancer. Thus, future studies of these human orthologs may identify potent tumor suppressors in cancer.

## Potential function of Non-stop and the NIC

Our proteomic, biochemical, and genetic analyses established that to maintain identity Non-stop acts as part of the NIC protein complex.

Like SAGA, NIC contained the entire DUB module including Non-stop, E(y)two and Sgf11. Within SAGA, Sgf11 is critical for Non-stop activity. However, loss of Sgf11 did not result in loss of EC identity. While this may reflect intrinsic differences between the complexes it may also be possible that knock-down of Sgf11 transcripts in ECs did not reduce Sgf11 protein levels sufficiently enough to result in a phenotype.

The BTB domain of Mod(mdg4) interacts with the M domain of Cp190 (*Melnikova et al., 2017*), and together they enable the recruitment of NIC to the promoter/enhancer regions. There are more than 30 Mod(mdg4) isoforms with different C-ends, each of which interacts with different DNA-binding TF (*Büchner et al., 2000*; *Melnikova et al., 2017*). Cp190 can also be able to recruit different TFs and interacts with Chromo and Zinc-finger 4 (Maksimenko, and Georgiev unpublished data), that are co-localized with Nup98-96. The Mod(mdg4)/CP190 sub-complex is bound to promoters through interactions with many promoter specific C2H2 proteins including dCTCF, Su(Hw)(ref), and possibly via other TFs enriched in our ATAC-seq analysis.

Prominent phenotypes of loss of Non-stop were the reduced protein level of Nup98 and its misslocalization from the nuclear periphery to an intranuclear punctate distribution. Nup98 was shown to recruit Set1~COMPASS to enhance histone H3K4me2-3 methylations in hematopoietic progenitors (*Franks et al., 2017*). Thus, a possible function of the NIC may be the recruitment of the H3K4me2/3 COMPASS methylases to catalyze H3K9 di- and tri-methylation at enhancers and promotors, which are fundamental for gene activation (*Nakanishi et al., 2009*; *Shilatifard, 2012*; *Sze and Shilatifard, 2016*). Members of the NIC complex were previously shown to form a Nuclear pore complex that enhances transcriptional memory upon exposure to ecdysone (*Pascual-Garcia et al., 2017*), raising the possibility that NIC is required for enhancing transcriptional memory promoting the transcription of EC-related genes and prevents Polycomb-dependent transcriptional repression.

In addition to NIC subunits the complex described by *Pascual-Garcia et al. (2017)* contained Thrithorax-like protein (Trl)/GAF and the boundary/architectural proteins CTCF, Suppressor of Hairy Wing (SuHw) and Mtor, but not Non-stop or E(y)2. Interestingly, the DNA binding site of Trl/GAF/ was enriched in EC genes that their expression required Non-stop for maintaining chromatin accessibility. However, Trl, CTCF, SuHw, or Mtor that interact with Cp190 were not detected as Non-stop bound proteins in our proteomic analysis, and RNAi-mediated elimination of Trl/GAF, or SuHw from ECs did not result in loss of EC identity. Thus, suggesting that these are likely separate regulatory complexes with shared subunits.

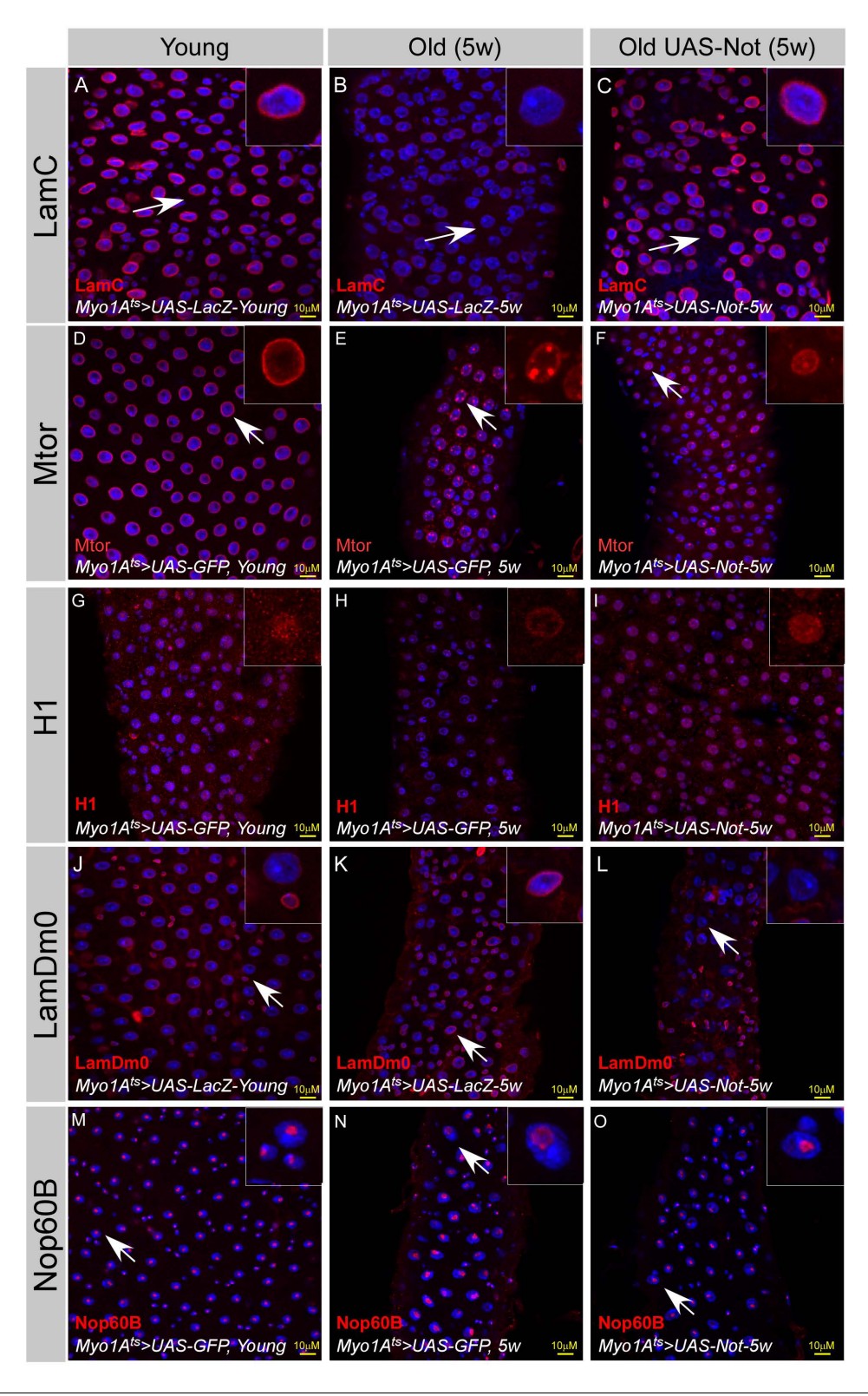

**Figure 8.** Expression of Non-stop restore large-scale organization of aged ECs. (A–O) Confocal images of the midgut tissue using the indicated antibodies and expressing the indicated transgenes in EC using the MyoIA-Gal4/Gal80ts. DAPI marks DNA, and scale bar is 10 μM. (A, D, G, J, M) Young Guts expressing UAS-LacZ. (B, E, H, K, N) Five weeks old guts expressing control (UAS-GFP). (C, F, I, L, O) Five weeks old guts expressing UAS-Non-stop. (A–C) α-LamC; (D–F) Mtor; (G–I) α-Histone H1; (J–L) α-LamDm0; (M–O) α-Nop60B.

*Figure 8 continued on next page*

*Figure 8 continued*

The online version of this article includes the following figure supplement(s) for figure 8:

**Figure supplement 1.** Expression of Non-stop restore large-scale organization of aged ECs.

We found that Non-stop is required for the stability of NIC in ECs. RNA-seq analysis established that Non-stop did not regulate the mRNA level of NIC subunits. Suggesting that Nonstop acts by deubiquitinating NIC subunits in ECs. In this regard, it is possible that the stabilization of NIC subunits may also require the EC-related lamin, LamC. We noticed that Lamin C protein levels (but not mRNA) decline upon loss of Non-stop. Moreover, LamC expression partially restored the protein levels of NIC subunits and their intranuclear localization, potentially by serving as a scaffold for NIC at the nuclear periphery (not shown). Therefore, additional experiments are required to establish whether NIC subunits are directly deubiquitinated and stabilized by Non-stop iso-peptidase activity and the potential contribution of LamC to stabilization of NIC subunits.

Thus, Non-stop may function at two levels; One is a direct role in transcription safeguarding chromatin accessibility at EC genes, while a second function is the stabilization of identity supervisors including NIC subunits and LamC.

## Crosstalk between identity supervisors;

Both Non-stop and the transcription factor Hey are bona-fide regulators of EC identity required for the expression of EC-related genes. A significant number of EC-related genes required both Non-stop and Hey for their expression, suggesting that Hey and Non-stop may co-regulate these genes. However, functional and epistatic tests suggest that Hey also acts upstream or in additional pathways to Non-stop. Hey binds to enhancers in *lamin* genes repressing the expression of the ISC-related lamin, LamDm0 and enhances the expression of LamC. In contrast, Non-stop does not regulate the accessibility, or expression of either LamDm0 or LamC at mRNA level. However, Non-stop is required for maintaining the levels of LamC protein. Therefore, loss of Non-stop results in a decline in LamC but not in the ectopic expression of LamDm0, which is observed upon acute loss of Hey or aging. This discrepancy may be due to the presence of Hey on its repressed targets in young ECs where Non-stop is targeted, and directly repressing their expression as maybe in the case of LamDm0. Moreover, EC-specific expression of Non-stop did not suppress the phenotypes associated with acute loss of Hey in young ECs further supporting for Hey-dependent, but Non-stop independent functions. However, the ECs-specific expression of either Non-stop or Hey in aging midguts restores expression of LamC and repressed ectopic LamDm0 expression.

## Non-stop and pre-mature aging

Changes in large-scale nuclear organization are hallmarks of aging (*Zhang et al., 2020*). Expression of identity supervisors can prevent age-related distortion of the nucleus EC identity and protect overall the epithelial tissue (This work and *Flint Brodsly et al., 2019*). However, to accomplish this, Non-stop or Hey were continuously expressed in ECs and temporal expression of Hey or Non-stop in already aged ECs was not sufficient to suppress aging phenotypes. Thus, if the levels of identity supervisors are kept at youthful levels, they can continue to maintain cell identity and prevent signs of aging, effectively keeping the gut organization and structure similar to young tissue (*Kenyon, 2010*).

Furthermore, it is not clear how expression of a single regulator like Non-stop has an extensive impact on the entire nucleus. Recent studies suggest that Non-stop functions in additional multiprotein complexes that may regulate large-scale cellular organization. For example, Non-stop is part of an Arp2/3 and WAVE regulatory (WRC) actin-cytoskeleton organization complex where it deubiquitinates the subunit SCAR (*Cloud et al., 2019*). In this regard, a nuclear actin organizing complex, WASH, interacted with nuclear Lamin and was required for large scale nuclear organization (*Verboon et al., 2015*). Thus, it is tempting to suggest that such complexes are required to maintain cell identity, and that subunits within these complexes are deubiquitinated by Non-stop.

In this regard many nuclear proteins are extremely long-lived proteins (LLPs) among them are nuclear pore complex proteins (NPCs) and core histones (*Toyama et al., 2013*; *Toyama et al., 2019*). The extended stability of LLPs may originate from intrinsic properties of LLPs, or due to

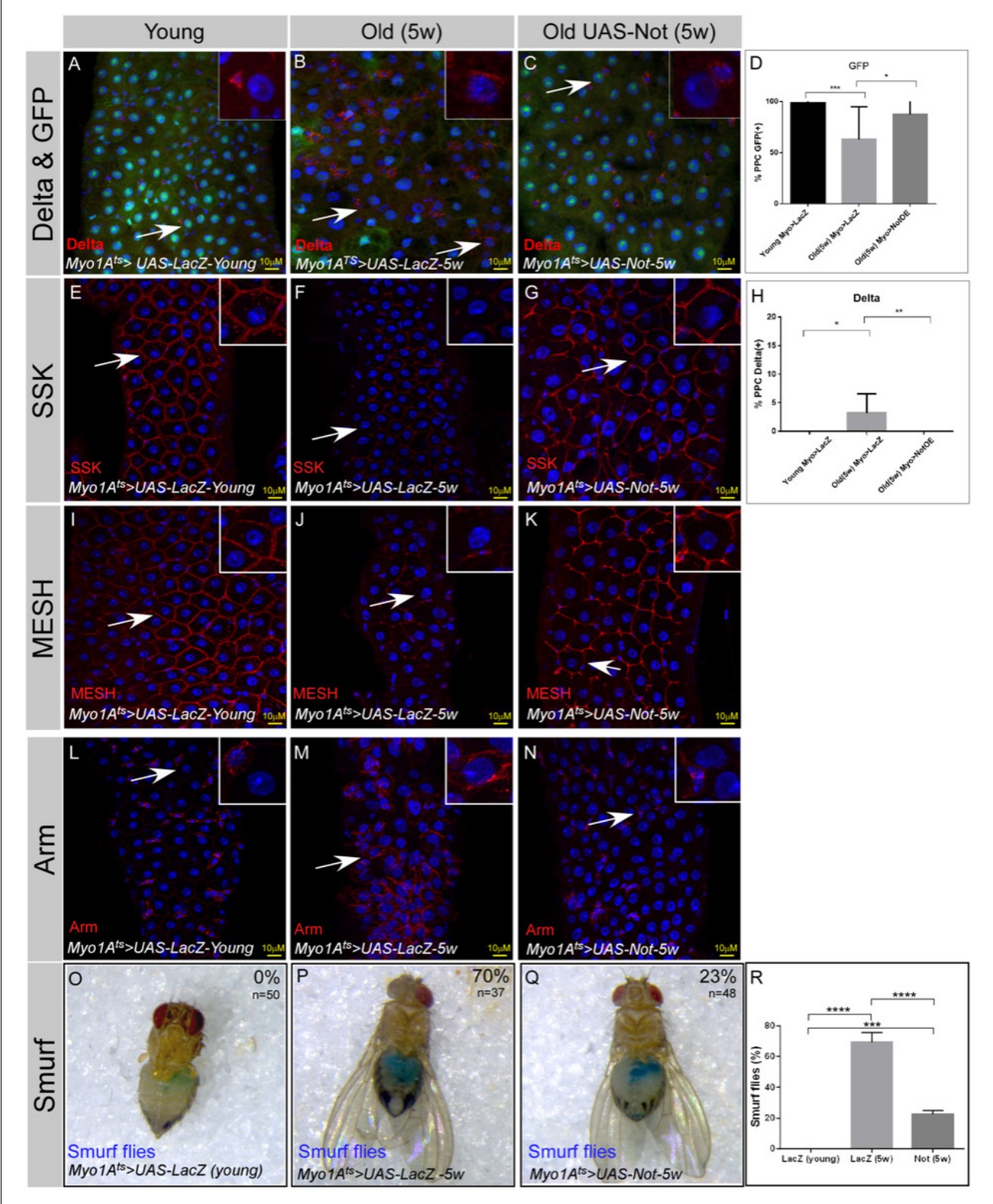

**Figure 9.** Continuous Expression of Non-stop in ECs suppresses aging phenotypes in the midgut. (A–N) Confocal images of the midgut tissue using the indicated antibodies and expressing the indicated transgenes in EC using the MyoIA-Gal4/Gal80ts. DAPI marks DNA and scale bar is 10 μM. (A, E, I, L) midguts derived from 2 to 4 days old flies (young) (B, F, J, M) Midguts derived from 5 weeks old flies expressing control (UAS-LacZ). (C, G, K, N) Midguts derived from 5 weeks old flies expressing UAS-Non-stop. DAPI marks DNA (blue), and scale bar is 10 μM. Arm, Armadillo; SSK Snakeskin. (D,

*Figure 9 continued on next page*

*Figure 9 continued*

H) Quantification of similar experiments shown in (A–C). **** = P < 0.0001, ***P < 0.001; **P<0.01; *=P<0.1 (O-R) Aging impairs gut integrity as evident by the leakage of blue-colored food into the abdomen (smurf assay). Continuous expression of Non-stop but not control using the MyoIA-Gal4/Gal80ts for five weeks safeguards gut integrity, n = 48, 38 respectively; P < 0.001.

The online version of this article includes the following source data for figure 9:

**Source data 1.** Quantification of cell populations described in 9D,H and Smurf assay in Figure 99R.

sequestration and evading degradation. However, increased stability may be also maintained by constitutive de-ubiquitination. Indeed, post-translational modification by ubiquitin and SUMO were shown to regulate lamin stability and their intranuclear localization (*Blank, 2020*). Specifically, type-A lamin and its splice variant Progerin, the cause of Hutchinson Gilford progeria syndrome (HGPS), a premature aging syndrome, are degraded by the HECT-type E3 ligase Smurf2 via ubiquitin-dependent autophagy (*Borroni et al., 2018*). The elimination of Progerin by expression of Smurf2 in HGSP-fibroblasts reduced the deformation observed in these cells. Thus, it is possible that enhancing Progerin degradation by inhibiting the human ortholog of Non-stop, USP22, will restore nuclear architecture, and suppress the premature aging phenotypes observed in HGPS cells.

## Materials and methods

- Key resource table with fly stocks and antibodies used in this study
- Plasmids and Primers used in this study
- Chemicals used

### Key resources table

| Reagent type (species) or resource | Designation | Source or reference | Identifiers | Additional information |
|---|---|---|---|---|
| Genetic reagent (*D. melanogaster*) | w; MyoIA-Gal4; tub-Gal80ts, UAS-GFP | Edgar Bruce lab | | |
| Genetic reagent (*D. melanogaster*) | w; esg-Gal4, tub-Gal80ts UAS-GFP | Edgar Bruce lab | | |
| Genetic reagent (*D. melanogaster*) | w; Prospero-Gal4 | Edgar Bruce lab | | |
| Genetic reagent (*D. melanogaster*) | w; Dl-Gal4/TM6, Tb | Edgar Bruce lab | | |
| Genetic reagent (*D. melanogaster*) | Su(H)-Gal4 | Sarah Bray lab | | |
| Genetic reagent (*D. melanogaster*) | Notch-reporter 3.37-gh-LacZ | Sarah Bray lab | | |
| Genetic reagent (*D. melanogaster*) | M5-4::LacZ | Erika Matunis | | |
| Genetic reagent (*D. melanogaster*) | UAS-LamC | Lori Walworth | | |
| Genetic reagent (*D. melanogaster*) | UAS-Non-stop RNAi | Bloomington | 28725 | |

*Continued on next page*

*Continued*

| Reagent type (species) or resource | Designation | Source or reference | Identifiers | Additional information |
|---|---|---|---|---|
| Genetic reagent (*D. melanogaster*) | UAS-Non-stop RNAi | VDRC | 45775/GD ; 45776/GD | |
| Genetic reagent (*D. melanogaster*) | UAS-GFP | Bloomington | # 1521 | |
| Genetic reagent (*D. melanogaster*) | UAS-GCN5-RNAi | Bloomington | # 33981 | |
| Genetic reagent (*D. melanogaster*) | UAS-SGF11-RNAi | VDRC | 17166/GD;100581/KK | |
| Genetic reagent (*D. melanogaster*) | UAS-Su(Hw)-RNAi | Bloomington | # 33906 | |
| Genetic reagent (*D. melanogaster*) | UAS-LacZ | Bloomington | #1776 | |
| Genetic reagent (*D. melanogaster*) | UAS-Atx7-RNA-i | VDRC | 102078/KK | |
| Genetic reagent (*D. melanogaster*) | UAS-e(y)2 RNAi | VDRC | 16751/GD ; 108212/KK | |
| Genetic reagent (*D. melanogaster*) | UAS-e(y)2 RNAi | Bloomington | 42524 | |
| Genetic reagent (*D. melanogaster*) | UAS-CP-190 RNAi | Bloomington | 42536 | |
| Genetic reagent (*D. melanogaster*) | UAS-Nup98-96 RNAi | Bloomington | #28562 | |
| Genetic reagent (*D. melanogaster*) | UAS-mod(mdg4) RNAi | Bloomington | # 32995 | |
| Genetic reagent (*D. melanogaster*) | "G-TRACE" (w*; P{UAS-RedStinger}6, P{UAS-FLP.Exel}3, P{Ubi-p63E(FRT.STOP)Stinger}15F2.) | Bloomington | #28281 | |
| Genetic reagent (Yeast) | pJ69-4A (MATa trp1-901 leu2-3,112 ura3-52 his3-200 gal4Δ gal80Δ GAL2-ADE2 LYS2::GAL1-HIS3 met2::GAL7-lacZ) | | | |
| Antibody | anti-Prospero (Mouse monoclonal (IgG1)) | DHSB | Prospero (MR1A) | (1:100) |
| Antibody | anti-Armadillo (Mouse monoclonal) | DHSB | N2 7A1 Armadillo | (1:500) |
| Antibody | anti-Delta (Mouse monoclonal(IgG1)) | DHSB | C594.9B | (1:50) |
| Antibody | anti 4F3 anti-discs large (Dlg) (Mouse monoclonal) | DHSB | 4F3 anti-discs | (1:50) |
| Antibody | anti-HP1 (Rabbit polyclonal) | Susan Purkhurst lab | | (1:1000) |

*Continued on next page*

*Continued*

| Reagent type (species) or resource | Designation | Source or reference | Identifiers | Additional information |
|---|---|---|---|---|
| Antibody | anti-mTor (Mouse monoclonal(IgG1)) | DHSB | 12F10-5F11 | (1:100) |
| Antibody | anti-βGal (Rabbit polyclonal) | MP Biomedicals | 55976 | (1:500) |
| Antibody | anti-Actin (Mouse monoclonal) | MP Biomedicals | 691001 | (WB) (1:4000) |
| Antibody | anti-MESH (Rabbit polyclonal) | Mikio Furuse lab | | (1:100) |
| Antibody | anti-SSK (Rabbit polyclonal) | Mikio Furuse lab | | (1:100) |
| Antibody | anti-caudal (Guinea Pig Polyclonal) | Jeff Reinitz lab | | (1:200) |
| Antibody | anti-odd-skipped (Guinea Pig Polyclonal) (Rat Polyclonal) | Jeff Reinitz lab | | (1:100) |
| Antibody | anti Lamin C (Mouse monoclonal) | Yossef Gruenbaum lab | | (1:500) |
| Antibody | anti Otefin (Mouse monoclonal) | Yossef Gruenbaum lab | | (1:10) |
| Antibody | anti-Lamin Dm0 (Rabbit polyclonal) | Yossef Gruenbaum lab | | (1:300) |
| Antibody | anti-p-histone H3 (Rabbit polyclonal) | Abcam | ab5176 | (1:100) |
| Antibody | anti-Nop60B (Rabbit polyclonal) | Steven Pole lab | | (1:100) |
| Antibody | Guinee pig anti-Coilin | Joseph Gall lab | | (1:2000) |
| Antibody | anti-Non-stop (Rabbit polyclonal) | *Cloud et al., 2019* | | (1:100) |
| Antibody | anti-PCNA (Rabbit polyclonal) | Bruce Edgar Lab | | (1:100) |
| Antibody | anti-Nup98 (Rabbit polyclonal) | Cordula Schlutz Lab | | (1:100) |
| Antibody | anti-e(y)2 (Rabbit polyclonal) | *Cloud et al., 2019* | | (1:1000 - WB, 1:100 - IHC) |
| Antibody | anti-Cp190 (Rabbit polyclonal) | *Golovnin et al., 2007* | | (1:1000WB), (1:100 IHC) |
| Antibody | anti-mod (MDG4) (Mouse monoclonal) | *Golovnin et al., 2007* | | ( 1:100 IHC) |
| Antibody | anti-H1 (Rabbit polyclonal) | Bas Van-Steensel lab | | (1:500) (1:1000WB) |
| Antibody | anti-H2Bub (Mouse monoclonal) | Moshe Oren Lab | | (1:500) |
| Antibody | anti-H2B (Mouse monoclonal) | Moshe Oren Lab | | (1:500) |
| Antibody | anti-Pdm1 (Rabbit polyclonal) | Di´az-Benjumea lab | | (1:50) |
| Antibody | Alexa Fluor 568 goat anti-mouse IgG1(γ1) | invitrogen | A21124 | (1:1000) |
| Antibody | Alexa Fluor 568 goat anti-mouse IgG (H+L) | invitrogen | A11031 | (1:1000) |
| Antibody | Alexa Fluor 568 goat anti-rabbit IgG (H+L) | invitrogen | A11036 | (1:1000) |
| Antibody | Alexa Fluor 633 goat anti-rabbit IgG (H+L) | invitrogen | A-21070 | (1:1000) |
| Antibody | Alexa Fluor 633 goat anti-mouse IgG1 (γ1) | invitrogen | A-21126 | (1:1000) |
| Antibody | Alexa Fluor 568 goat anti-guinea pig | invitrogen | A11075 | (1:1000) |
| Antibody | Alexa Fluor 633 goat anti-guinea pig | invitrogen | A21105 | (1:1000) |

*Continued on next page*

*Continued*

| Reagent type (species) or resource | Designation | Source or reference | Identifiers | Additional information |
|---|---|---|---|---|
| Antibody | Alexa Fluor 633 goat anti-rat | invitrogen | A21094 | (1:1000) |
| Antibody | Alexa Fluor 568 goat anti-rat | invitrogen | A11077 | (1:1000) |
| sequence-based reagent | CP190 CT (aa, 468-1096) | pET32a(+) vector (merck) | | 5'-tttggtaccgggccctggctgtgcctg-3' |
| Sequence-based reagent | CP190 CT (aa, 468-1096) | pET32a(+) vector (merck) | | 5'-tttctcgagtgcggccgcagatcttag-3' |
| Sequence-based reagent | CP190 NT (aa, 1-524) | pET32a(+) vector (merck) | | 5'- tttcatatgggtgaagtcaagtccgtg -3' |
| Sequence-based reagent | CP190 NT (aa, 1-524) | pET32a(+) vector (merck) | | 5'- tttctcgagcatgtggaaatgcagttcccg -3' |
| Sequence-based reagent | e(y)2 | pET32a(+) vector (merck) | | 5'- tttggatccccggaattcccgacgatgag-3' |
| Sequence-based reagent | e(y)2 | pET32a(+) vector (merck) | | 5'- tttgcggccgcttaggattcgtcctctggc-3' |
| Sequence-based reagent | Non-Stop (aa 496) | pGBT9 vector (Clontech) | | 5'-ttgaattcatgtccgagacgggttgtc-3' |
| Sequence-based reagent | Non-Stop (aa 496) | pGBT9 vector (Clontech) | | 5'-ttgtcgacttactcgtattccagcacatt-3' |
| Sequence-based reagent | CP190 (aa 1096) | pGAD424 vector (Clontech) | | 5'-ttcccgggcatgggtgaagtcaagtccg-3' |
| Sequence-based reagent | CP190 (aa 1096) | pGAD424 vector (Clontech) | | 5'-tttggaggagctatatttactaagatct-3' |
| Sequence-based reagent | CP190 from first to fourth zinc fingers | pGAD424 vector (Clontech) | | 5'-ttgaattcgagaatactactgggccct-3' |
| Sequence-based reagent | CP190 from first to fourth zinc fingers | pGAD424 vector (Clontech) | | 5'-ttgtcgacgccatcctccaaagcctg-3' |
| Sequence-based reagent | CP190 from second to third | pGAD424 vector (Clontech) | | 5'-ttgaattcgcgctttgtgagcattgc-3' |
| Sequence-based reagent | CP190 from second to third | pGAD424 vector (Clontech) | | 5'-ttgtcgacgttgtcgtccgtgtgcac-3' |
| Sequence-based reagent | CP190Δ4 | pGAD424 vector (Clontech) | | 5'-aaggtaccggagcaggctttgga-3' |
| Sequence-based reagent | CP190Δ4 | pGAD424 vector (Clontech) | | 5'-aaggtacccactgctgcttgttgtcg-3' |
| Sequence-based reagent | CP190Δ3-4 | pGAD424 vector (Clontech) | | 5'-aaggtaccggagcaggctttgga |
| Sequence-based reagent | CP190Δ3-4 | pGAD424 vector (Clontech) | | 5'-aaggtaccaacgtatacagcagcgac-3' |
| Sequence-based reagent | CP190Δ2-4 | pGAD424 vector (Clontech) | | 5'-aaggtaccggagcaggctttgga |
| Sequence-based reagent | CP190Δ2-4 | pGAD424 vector (Clontech) | | 5'-aaggtacccgcgccggatcaattg-3' |
| Sequence-based reagent | CP190Δ1-4 | pGAD424 vector (Clontech) | | 5'-gccctggctgaaggagcaggctttggagga |
| Sequence-based reagent | CP190Δ1-4 | pGAD424 vector (Clontech) | | 5'-cctgctccttcagccagggcccagtagtat-3' |
| Cell line (*D. melanogaster*) | S2 | | | |
| recombinant DNA reagent | pY3H | | | |

*Continued on next page*

*Continued*

| Reagent type (species) or resource | Designation | Source or reference | Identifiers | Additional information |
|---|---|---|---|---|
| recombinant DNA reagent | pRmha3 C-HAx2-FLx2-nonstop-735 | *Cloud et al., 2019* | | |
| Transfected construct (*Drosophila* S2 Cells) | Non-Stop-RNAi forward | | | 5'-cggaattccgaattaatacgactca ctataggggatttaatctggaaccatgcgaa-3' |
| Transfected construct (*Drosophila* S2 Cells) | Non-Stop-RNAi reverse | | | 5'-cggaattccgaattaatacgactc actatagggaaatgtcccaaaacggatcgta-3' |
| Chemical compound, Drug | Diamidino-2-phenylindole* dihydrochl [DAPI] 1mg | Sigma | D9542-1MG | 1:1000 |
| Chemical compound, Drug | Bromophenol Blue | Sigma | #B5525 | |
| Chemical compound, Drug | Guanidine hydrochloride | Sigma | #G4505 | |
| Chemical compound, Drug | NP40 (Igepal CA-630) | Sigma | #I3021 | |
| Chemical compound, Drug | Triton X-100 | Amresco | #0694 | |
| Chemical compound, Drug | Acrylamide (Bis-Acrylamide 29:1) | Biological Industries | #01-874-1A | |
| Chemical compound, Drug | Ammonium Persulfate | Sigma | #A-9164 | |
| Chemical compound, Drug | TEMED | Sigma | #T-7024 | |
| Chemical compound, Drug | L-Glutamine | Gibco | #25030024 | |
| Chemical compound, Drug | MG132 | Boston Biochemicals | | |
| Chemical compound, Drug | Blot Qualified BSA | Biological Industries | #PRW3841 | |
| Chemical compound, Drug | Agarose | SeaKem LE Agarose-Cambrex Bio Science | #CAM-50004 | |
| Chemical compound, Drug | Bradford Protein Assay | BioRad | #500-0006 | |
| Chemical compound, Drug | EZ-ECL | Biological Industries | #20-500-500 | |
| Chemical compound, Drug | FD&C blue dye #1 | | | |
| Chemical compound, Drug | Cyclohexamide | Sigma | #01810 | |

## Methods

- In vitro binding
- Direct Yeast 2 Hybrid
- Proteomic analysis of Non-stop associated proteins
- RNAi in *Drosophila* S2 cells
- Conditional expression of transgenes in specific gut cells
- Conditional G-TRACE analysis
- Gut dissection and immunofluorescence detection
- Gut integrity and tracing of organismal survival
- Genomic analysis; RNA-seq, ATAC-seq and bioinformatics analyses including RNA extraction, cDNA preparation and Gene expression and RNA-seq, and bioinformatics analyses.
- Statistical analysis

## Fly stocks used in this study

Fly stocks were maintained on yeast-cornmeal-molasses-malt extract medium at 18°C or as stated in the text. UAS- RNAi used in the screen are described under *Figure 1—source data 2*.

## UAS and Gal4 transgenic lines used

All transgenic RNAi lines used for the Ub/Ubl screen are detailed in *Figure 1—source data 2*. All other lines used in this study are described in the Key resources table.

## Antibodies used in this study

All primary and secondary antibodies used are described in the Key resource table.

## Plasmids and primers

pRmha3 C-HAx2-FLx2-nonstop-735 – was as described in *Cloud et al., 2019*.

### Plasmids for in vitro binding

CP190 CT (aa, 468–1096) was PCR-amplified using primers 5′-tttggtaccgggccctggctgtgcctg-3′ and 5′-tttctcgagtgcggccgcagatcttag-3′ and subcloned into pET32a(+) vector (Merck Biosciences) in frame with 6xHis tag using restriction sites *Kpn*I and *Xho*I.

CP190 NT (aa, 1–524) was PCR-amplified using primers 5′- tttcatatgggtgaagtcaagtccgtg −3′ and 5′- tttctcgagcatgtggaaatgcagttcccg −3′ and subcloned into pET32a(+) vector (Merck Biosciences) in frame with 6xHis tag using restriction sites *Nde*I and *Xho*I.

E(y)two was PCR-amplified using primers 5′- tttggatccccggaattcccgacgatgag-3′ and 5′-tttgcggccgcttaggattcgtcctctggc-3′ and subcloned into pET32a(+) vector (Merck Biosciences) in frame with 6xHis tag using restriction sites *Bam*HI and *Not*I.

### Plasmids used in the yeast two-hybrid assay

The full-sized Non-stop (aa 496) was PCR-amplified using primers 5′-ttgaattcatgtccgagacgggttgtc-3′ and 5′-ttgtcgacttactcgtattccagcacatt-3′ and subcloned into pGBT9 vector (Clontech) in frame with DNA-binding domain of GAL4 using restriction sites *Eco*RI and *Sal*I.

For Y3H and Y4H assays we used plasmid pY3H and corresponding cDNAs were subcloned using the same restriction sites. We assembled the quadruple complex by creating a plasmid that encoded a fusion protein Eny2-Sgf11separated by a self-cleaving 2A peptide (*Souza-Moreira et al., 2018*). Primers for cloning of Eny2-2A-Sgf11 product: 5′-tctttgttgaaattggctggtgatgttgaattgaatccaggtc-caatgtctgcagccaacatgc-3′ and 5′-accagccaatttcaacaaagaaaaattagtagcaccaccagaaccg-gattcgtcctctggctc-3′.

The full-sized CP190 (aa 1096) was PCR-amplified using primers 5′-ttcccgggcatgggtgaagt-caagtccg-3′ and 5′-tttggaggagctatatttactaagatct-3′ and subcloned into pGAD424 vector (Clontech) in frame with activation domain of GAL4 using restriction sites *Sma*I and *Bam*HI. Fragments of CP190 from first to fourth zinc fingers was PCR-amplified using primers 5′-ttgaattcgagaatac-tactgggccct-3′ and 5′-ttgtcgacgccatcctccaaagcctg-3′, from second to third - 5′-

ttgaattcgcgctttgtgagcattgc-3' and 5'-ttgtcgacgttgtcgtccgtgtgcac-3' and then subcloned into pGAD424 vector (Clontech) in frame with activation domain of GAL4 using restriction sites *Eco*R1 and *Sal*1.

Corresponding primers were used to make full-sized deletion variants of CP190:

CP190Δ4 5'-aaggtaccggagcaggctttgga-3' and 5'-aaggtacccactgctgcttgttgtcg-3';
CP190Δ3-4 5'-aaggtaccggagcaggctttgga and 5'-aaggtaccaacgtatacagcagcgac-3';
CP190Δ2-4 5'-aaggtaccggagcaggctttgga and 5'-aaggtacccgcgccggatcaattg-3';
CP190Δ1-4 5'-gccctggctgaaggagcaggctttggagga and 5'-cctgctccttcagccagggcccagtagtat-3':

## Primers used for Non-stop RNAi in *Drosophila* cells

- Non-stop-RNAi forward – 5'-cggaattccgaattaatacgactcactatagggatttaatctggaaccatgcgaa-3'
- Non-stop-RNAi reverse – 5'-cggaattccgaattaatacgactcactatagggaaatgtcccaaaacggatcgta-3'

## Chemicals

Bromophenol Blue (Sigma #B5525), Guanidine hydrochloride (Sigma #G4505), NP40 (Ipegal CA-630) (Sigma #I3021), Triton X-100 (Amresco #0694), Acrylamide (Bis-Acrylamide 29:1) (Biological Industries #01-874-1A), Ammonium Persulfate (Sigma #A-9164), TEMED (Sigma #T-7024), L-Glutamine (Gibco #25030024), MG132 (Boston Biochemicals), Blot Qualified BSA (Biological Industries #PRW3841), Agarose (SeaKem LE Agarose- Cambrex Bio Science #CAM-50004), Bradford Protein Assay (BioRad #500–0006), EZ-ECL (Biological Industries #20-500-500), FD and C blue dye #1, Cyclohexamide (Sigma #01810).

## Methods

### Co-immunoprecipitation

Whole cell extracts were prepared from OregonR flies by grinding with a Teflon pestle in Extraction Buffer (20 mM HEPES (pH7.5), 25% glycerol,1.5 mM MgCl$_2$, 420 mM NaCl, 0.2 mM EDTA) with 1% NP-40 and protease inhibitors. Heads were grinded with a pestle until the solution was homogenous. Insoluble proteins were separated by centrifugation for 30 min at 4°C at 15,000 *g*. The supernatant was placed into a new tube and half volume of Dignum A buffer (20 mM HEPES (pH7.5),1.5 mM MgCl$_2$, 10 mM KCl) was added to the lysates in order to adjust the salt concentration. Centrifugation was repeated. The supernatant was placed into a new tube and anti-Non-stop primary antibody was added in a dilution 1:25 and then rotated at 4°C for 4 hr. After 4 hr, the supernatant was then added to Dynabeads Protein A (Sigma- Aldrich Catalog number: 10002D) and rotated at 4°C for 2 hr. After 2 hr, the supernatant was removed and stored then the Dynabeads were washed five times in extraction buffer plus Dignum A. The Dynabeads were resuspended in 30 µl 2X Urea and boiled for 5 min at 95°C to release immunoprecipitated proteins.

### In vitro binding

6xHis-tagged proteins were expressed and purification from *E. coli* BL-21 (DE3), using Ni-NTA agarose beads. His-tagged protein were induced with 0.5 mM IPTG for 5 hr at 30°C and subsequently immobilized on with Ni-NTA agarose beads. Nuclear extract derived from Non-stop expressing S2 cells was prepared similar to the described in *Cloud et al., 2019*, see 'Non-denaturing extract': Stably transfected cells were resuspended in Extraction Buffer (20 mM HEPES (pH7.5), 25% Glycerol, 420 mM NaCl, 1.5 mM MgCl$_2$, 0.2 mM EDTA, 1:100 ethidium bromide with protease inhibitors added). 1% NP-40 was added and the cells were pipetted up and down until the solution was homogenous. They were placed on ice for one hour with agitation every 10–15 min. They were then centrifuged for 30 min at 4°C at 20,000 x *g*. An equal volume of Dignum A buffer (10 mM HEPES (pH 7.5), 1.5 mM MgCl$_2$, 10 mM KCl) was added to the lysates in order to adjust the salt concentration to 210 mM NaCl.

Binding was performed using 0.5 mg of S2 cell extract expressing HF-Non-stop and the indicated His-tagged proteins immobilized to Ni-NTA beads using binding buffer (20 mM Hepes-KOH pH 7.7, 150 mM NaCl, 10 mM MgCl$_2$0.1%mM ZnCl$_2$, 0.1% NP40, 10% Glycerol and protease inhibitors) for

over-night in rotation. Subsequently, beads were collected and washed four times with wash buffer, and proteins resolved over SDS-PAGE and detected by western blot analysis.

## Yeast two-hybrid assays (Y2H, Y3H, Y4H)

Y2H was carried out using yeast strain pJ69-4A (MATa trp1-901 leu2-3,112 ura3-52 his3-200 gal4Δ gal80Δ GAL2-ADE2 LYS2::GAL1-HIS3 met2::GAL7-lacZ), with plasmids according to Clontech protocols . In brief, for growth assays, AD (activation domain of GAL4) - and BD (DNA-binding domain of GAL4) -fused plasmids were co-transformed into yeast strain pJ69-4A by the lithium acetate method, as described by the manufacturer with some modifications. For Y3H (Yeast three-hybrid assay) and Y4H (Yeast four-hybrid assay) pY3H plasmid was used for co-transformation. The assembly and detection of the quaternary complex in yeast cells was carried out by the simultaneous expression of proteins fused with AD, BD, as well as a fusion protein product consisting of two proteins separated by an optimized 2A-peptide of equine rhinitis B virus 1 (ERBV-1), for which the indicator of the efficiency of self-cleavage in *Saccharomyces cerevisiae* cells is one of the highest among the known sequences (more than 90%) (*Souza-Moreira et al., 2018*). Transformed cells were plated on selective medium lacking Leu (leucine biosynthesis gene Leu2 is expressed from pGAD424 plasmid) and Trp (tryptophan biosynthesis gene Trp1 is expressed from pGBT9 plasmid) ('medium-2') for Y2H and lacking Leu, Trp and Ura (uracil biosynthesis gene Ura3 is expressed from pY3H plasmid) for Y3H/Y4H. The plates were incubated at 30°C for 2–3 days. Afterward, the colonies were streaked out on plates on selective medium lacking either (Leu, Trp and His) or (Leu, Trp, Ura and His) (histidine biosynthesis gene His3 is used as reporter) ('medium-3/–4'). The plates were incubated at 30°C for 3–4 days, and growth was assessed. The positive growth of yeast on selective 'medium-3/–4' indicates a physical interaction between tested protein molecules. Each assay was prepared as three independent biological replicates with three technical repeats.

## Proteomic analysis

Multidimensional protein identification technology and Mass spectrometry data processing were identical to the described in detail at *Cloud et al., 2019*; Multidimensional protein identification technology (MudPIT) and Mass spectrometry data processing were identical to that described in *Cloud et al., 2019*. *MudPIT*: TCA-precipitated protein pellets were solubilized using Tris-HCl pH 8.5 and 8 M urea, followed by addition of TCEP (Tris(2-carboxyethyl)phosphine hydrochloride; Pierce) and CAM (chloroacetamide; Sigma) were added to a final concentration of 5 mM and 10 mM, respectively. Proteins were digested using Endoproteinase Lys-C at 1:100 w/w (Roche) at 37°C overnight. Samples were brought to a final concentration of 2 M urea and 2 mM $CaCl_2$ and a second digestion was performed overnight at 37°C using trypsin (Roche) at 1:100 w/w. The reactions were stopped using formic acid (5% final). The digested size exclusion eluates were loaded on a split-triple-phase fused-silica micro-capillary column and placed in-line with a linear ion trap mass spectrometer (LTQ, Thermo Scientific), coupled with a Quaternary Agilent 1100 Series HPLC system. The digested Non-stop and control FLAG-IP eluates were analyzed on an LTQ-Orbitrap (Thermo) coupled to an Eksigent NanoLC-2D. In both cases, a fully automated 10-step chromatography run was carried out. Each full MS scan (400–1600 m/z) was followed by five data-dependent MS/MS scans. The number of the micro scans was set to one both for MS and MS/MS. The settings were as follows: repeat count 2; repeat duration 30 s; exclusion list size 500 and exclusion duration 120 s, while the minimum signal threshold was set to 100. *Mass Spectrometry Data Processing*: The MS/MS data set was searched using ProLuCID (v. 1.3.3) against a database consisting of the long (703 amino acids) isoform of non-stop, 22,006 non-redundant *Drosophila melanogaster* proteins (merged and deduplicated entries from GenBank release 6, FlyBase release 6.2,2 and NCI RefSeq release 88), 225 usual contaminants, and, to estimate false discovery rates (FDRs), 22,007 randomized amino acid sequences derived from each NR protein entry. To account for alkylation by CAM, 57 Da were added statically to the cysteine residues. To account for the oxidation of methionine to methionine sulfoxide, 16 Da were added as a differential modification to the methionine residue. Peptide/spectrum matches were sorted and selected to an FDR less than 5% at the peptide and protein levels, using DTASelect in combination with swallow, an in-house software.

The permanent URL to the dataset is: ftp://massive.ucsd.edu/MSV000082625. The data is also accessible from: ProteomeXChange accession: PXD010462 http://proteomecentral.

proteomexchange.org/cgi/GetDataset?ID=PXD010462. MassIVE | Accession ID: MSV000082625 - ProteomeXchange | Accession ID: PXD010462.

## RNAi in S2 cells

Verified S2 Schneider cells were obtained from DGRC (*Drosophila* Genomics Resource Center #181, RRID:CVCL_Z992) were maintained in Schneider's media supplemented with 10% fetal bovine serum and 1% penicillin-streptomycin (Thermo-Fisher, Catalog number: 15070063, 5000 U/ml) RNAi in S2 cells was performed as described in *Abed et al., 2011*.

## Conditional expression of transgenes in specific gut cells

Conditional expression of transgenic lines in specific midgut cells was achieved by activating a UAS-transgene under the expression of the cell-specific Gal4-drivers together with the tub-Gal80$^{ts}$ construct (*Jiang et al., 2009*). Flies were raised at 18°C. 2–4 days old, F1 adult progeny were transferred to the restrictive temperature 29°C (Gal80 off, Gal4 on) for two days unless indicated otherwise, dissected and analyzed. At least three biological independent repeats were performed for each experiment. Where possible, multiple RNAi lines were used.

## Conditional G-TRACE analysis

G-TRACE analyses was as described in *Flint Brodsly et al., 2019* using Myo-Gal4; G-TRACE flies were crossed to UAS-LacZ; Gal80$^{ts}$ (control) or UAS-Non-stop RNAi; Gal80$^{ts}$ and the appropriate genotypes were raised at 18°C (a temperature where no G-TRACE signal was detected). At 2–4 days, adult females were transferred to 29°C and linage tracing was performed.

## Gut dissection and immunofluorescence detection

Gut fixation and staining were carried out as previously described (*Shaw et al., 2010*; *Flint Brodsly et al., 2019*).

## Gut integrity and animal survival

Young female flies from the indicated genotype were collected into a fresh vial (10 flies per vial), that were kept in a humidified, temperature-controlled incubator at 29°C for the indicated time period. Smurf assay was performed as described in *Flint Brodsly et al., 2019*. Flies were transferred into vials containing fresh food every two days and were scored for viability at the indicated time points. LT50 (lethal time in days at which 50% of the flies died) analysis was calculated using the GraphPad Prism 5.00 (GraphPad Software, San Diego, CA, USA).

## Genomic studies

### RNA-sequencing

#### RNA Sample Preparation:

RNA-seq was performed similar to the described in *Flint Brodsly et al., 2019*. In brief, Adult *Drosophila* (2–4 days old) females, from four biological repeats, in which UAS-Non-stop RNAi or control UAS-GFP RNAi were expressed in ECs using MyoIAts and dissected in Ringer's solution on ice. The solution was then discarded, and the guts were disrupted by adding 350 µl RLT+β ME buffer (350 µl RLT+ 3.5 µl β-ME). Guts were than vortex for homogenization. 350 µl of 70% ethanol was then added and mixed well by pipetting. Guts were uploaded into RNeasy spin column and RNA purified according to the manufacture instructions. Sample quality (QC) Quality measurements for total RNA were performed using the TapeStation 2200 (Agilent).

### Library preparation and data generation of RNA-sequencing

Eight RNA-seq libraries were produced using the NEBNext Ultra Directional RNA Library Prep Kit for Illumina (NEB, cat no. E7420) according to manufacture protocol and starting with 100 ng of total RNA. mRNA pull-up was performed using the Magnetic Isolation Module (NEB, cat no. E7490). Two out of the twelve libraries (samples B1 and B2) were disqualified based on low library yield and high levels of adaptor dimer. The remaining ten libraries were mixed into a single tube at an equal molar concentration. The RNA-seq data was generated on two lanes of HiSeq2500, 50 SR.

NGS QC, alignment and counting 50 bp single-end reads were aligned to *Drosophila* reference genome and annotation file (*Drosophila melanogaster* BDGP6 downloaded from ENSEMBL) using TopHat (v2.0.13) allowing two mismatches per read with options -very-sensitive. The number of reads per gene was counted using Htseq (0.6.0).

### Descriptive and RNA-seq DEGs analysis

Samples' clustering and differential expressed genes (DEGs) were calculated using Deseq2 package (version 1.10.1). The similarity between samples was evaluated using correlation matrix, shown a heat plot and Principal Component Analysis (PCA). Samples belonging to the same group were more similar then samples from different experimental groups (*Figure 5—figure supplement 1A*). The expression ~12,000,000 fly transcripts were compared using DESeq2 and list of the differentially express genes (DEGs) was extracted into excel files. At adjusted p-value (p-adj) and lt;0.01 and LogFC and gt; one or LogFC and lt; −1, 1428 DEGs were found between guts derived from the control vs Non-stop RNAi targeted EC. Moreover, the expression of Non-stop was found to be downregulated by −0.88 log2FC between Non-stop RNAi vs. Ctrl with p-adj of less than $10^{-9}$.

### ATAC (Assay for Transposase-Accessible Chromatin) sequencing

Adult *Drosophila* (2–4 days old) females, from three biological repeats, in which Non-stop RNAi or GFP RNAi control were expressed in ECs using MyoIAts were dissected in ice cold Ringer's solution, and immediately placed in 25 µl of ice cold ATAC lysis buffer (10 mM Tris-HCl, pH 7.4,10 mM NaCl, 3 mM $MgCl_2$, 0.1% IGEPAL CA-630). Lysed guts were then centrifuged at 500x*g* for 15 min at 4'C and the supernatant was discarded. The rest of the ATAC-seq protocol was performed as described in *Buenrostro et al., 2013*. The final library was purified using a Qiagen MinElute kit (Qiagen) and Ampure XP beads (Ampure) (1:1.2 ratio) were used to remove remaining adapters. All samples were quantified using Qubit DNA HS assay. The final library was first checked on an Agilent Bioanalyzer 2000 for quality and the average fragment size. Successful libraries were sequenced with NextSeq 75 cycles high-output flow-cell, targeting ~25 million reads/sample.

### Bioinformatic analysis of ATAC-seq

Raw reads were trimmed for adapters and aligned to the *Drosophila melanogaster* reference genome using bowtie2. Redundant duplicated reads that aligned to the exact locations were removed from the aligned results, and then converted to the tagAlign format with consideration of the strand shift ('+" strand reads shifted by 4 bp, and '-" strand reads shifted by −5 bp). The tagAlign format alignment results were used to call peaks using MAC2. Narrow peaks with p<0.1 were reported. Peaks were compared across biological duplicates and pseudo duplicates (i.e., random subsets from a sample) to get IDR peaks that are supposed to be consensus across duplicates. Test for the difference between the two groups (control and Non-stop-RNAi) has been performed with the Bioconductor R package DiffBind. Significance is set by FDR < 0.1.

### Comparisons between RNA and ATAC sequencing data

The two data sets of DEGs and DBAs were analyzed to create a list of significant differentially bound peaks that are close to genes from the top 1428 DE genes. The thresholds that have been used to associate the peaks to genes is within 10 kb upstream and 10 kb downstream of the genes. To look for common binding motives between the two data sets, the genes were sub-divided into four categories: 1. peaks overlapping promoters. 2. peaks upstream of promoters. 3. peaks in genic region. 4. peaks downstream of genes. Next, I performed motif finding informatics using CONSENSUS, MDScan and MEME software.

Gene ontology analysis of mRNA expression and ATAC-seq was performed using the online softwares Metascape or STRING using default settings.

### Statistical analysis

Data was collected from three independent experiments. Statistical analysis, z-test comparisons were performed using Prism6 ANOVAs software. Significance is indicated by *** = p<0.001 and ** = p<0.01.

## Acknowledgements

We are grateful for mass spectrometry done by Skylar Martin-Brown, Laurence Florens, and Michael P Washburn at the Stowers Institute. We would like to thank Sarah Bray, Adi Salzberg, Bruce Edgar, Jeff Reinitz, Lori Wallrath, Pamela Geyer, Yossi Greenbaum, Lorry Pile, Erika Matunis, Dĭaz-Benjumea, Mikio Furuse Bas Van-Steensel, Moshe Oren, Joseph Gall, the Bloomington, VDRC, and NIG-FLY *Drosophila* stock centers for sharing antibodies, fly lines, reagents, and data. This research was supported by: The School of Biological and Chemical Sciences, UMKC; University of Missouri Research Board; UMKC SEARCH, UMKC SUROP scholars programs, and NIH Academic Development Via Applied and Cutting Edge Research (ADVANCER) program; NIGMS grant 5R35GM118068 to RM. Washington University School of Medicine / St. Louis Children's Hospital Children's Discovery Institute, MC-II-2014–363 to TD. Russian Science Foundation 19-74-30026 to PG, and the Israel Science Foundation (ISF) (Grants 739/15, 318/20), and by the Flinkman Marandi Family cancer research grant to AO.

## Additional information

### Funding

| Funder | Grant reference number | Author |
| --- | --- | --- |
| NIH NIGMS | 5R35GM118068 | Ryan D Mohan |
| CDI | MC-II-2014-363 | Todd Druley |
| Russian Science Foundation | 19-74-30026 | Pavel Georgiev |
| Israel Academy of Sciences and Humanities | 719/15 | Amir Orian |
| Israel Academy of Sciences and Humanities | 318/20 | Amir Orian |
| University of Missouri-Kansas City | | Ryan D Mohan |
| School of Biological Sciences, Washington State University | | Ryan D Mohan |
| University of Missouri-Kansas City | UMKC SEARCH | Ryan D Mohan |
| University of Missouri-Kansas City | UMKC SUROP scholars program | Ryan D Mohan |
| NIH | ADVANCER | Ryan D Mohan |
| Sydney West Translational Cancer Research Centre | | Amir Orian |
| Flinkman-Marandy cancer research grant | AO0001 | Amir Orian |

The funders had no role in study design, data collection and interpretation, or the decision to submit the work for publication.

### Author contributions

Neta Erez, Amir Orian, Conceptualization, Data curation, Formal analysis, Funding acquisition, Investigation, Writing - original draft, Project administration, Writing - review and editing; Lena Israitel, Conceptualization, Data curation, Formal analysis, Investigation, Writing - original draft, Writing - review and editing; Eliya Bitman-Lotan, Conceptualization, Data curation, Supervision, Funding acquisition, Investigation, Writing - original draft, Writing - review and editing; Wing H Wong, Gal Raz, Data curation, Formal analysis, Investigation; Dayanne V Cornelio-Parra, Salwa Danial, Na'ama Flint Brodsly, Elena Belova, Pavel Georgiev, Todd Druley, Conceptualization, Data curation, Formal analysis, Supervision, Funding acquisition, Investigation, Writing - original draft, Writing - review and editing; Oksana Maksimenko, Conceptualization, Data curation, Formal analysis, Supervision, Funding acquisition, Investigation; Ryan D Mohan, Conceptualization, Data curation, Formal analysis,

Supervision, Funding acquisition, Investigation, Writing - original draft, Project administration, Writing - review and editing

## Author ORCIDs
Ryan D Mohan https://orcid.org/0000-0002-7624-4605
Amir Orian https://orcid.org/0000-0002-8521-1661

## Decision letter and Author response
Decision letter https://doi.org/10.7554/eLife.62312.sa1
Author response https://doi.org/10.7554/eLife.62312.sa2

# Additional files
## Supplementary files
• Transparent reporting form

## Data availability
The following sequencing data were deposited: RNAseq and ATAC-seq data are available at NCBI through the Accession number PRJNA657899. The proteomic data set can be found at: ftp://massive.ucsd.edu/MSV000082625. The data are also accessible from ProteomeXChange and MassIVE (see Major datasets - generated).

The following datasets were generated:

| Author(s) | Year | Dataset title | Dataset URL | Database and Identifier |
|---|---|---|---|---|
| Erez O, Wong B | 2020 | A Non-stop identity complex (NIC) supervises enterocyte identity and protects from pre-mature aging | https://www.ncbi.nlm.nih.gov/bioproject/?term=PRJNA657899 | NCBI BioProject, PRJNA657899 |
| Florens L | 2019 | MudPIT analyses of the proteins co-purified with FLAG-HA Non-Stop FLAG-IPed from *Drosophila melanogaster* S2 cells | http://proteomecentral.proteomexchange.org/cgi/GetDataset?ID=PXD010462 | ProteomeXchange, PXD010462 |
| Florens L | 2019 | FTP directory/MSV000082625 at massive.ucsd.edu | ftp://massive.ucsd.edu/MSV000082625 | MassIVE, MSV000082625 |

The following previously published datasets were used:

| Author(s) | Year | Dataset title | Dataset URL | Database and Identifier |
|---|---|---|---|---|
| Expression profiling by array, Genome binding/occupancy profiling by genome tiling array, Ncbi Gene | 2016 | Expression profiling by array and Genome binding/occupancy profiling by genome tiling array. Ncbi gene | https://www.ncbi.nlm.nih.gov/geo/query/acc.cgi?acc=GSE87896 | NCBI Gene Expression Omnibus, GSE87896 |
| Orian A, Flint-Brodsly N, Bitman-Lotan E | 2018 | RNAseq analysis of whole Guts over expressing LaminDm0 or GFP in Enterocytes. NCBI Gene Expression Omnibus. | https://www.ncbi.nlm.nih.gov/geo/query/acc.cgi?acc=GSE112640 | NCBI Gene Expression Omnibus, GSE112640 |

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
