## [Decision Letter]

**Acceptance summary:**

The paper is of potential interest to a broad audience of biologists who are interested in homeostasis in the aging digestive system/gut. Using different approaches, the authors identify Non-stop (Not) functions in a complex to maintain the identity of Enterocyte cells and protects them from premature aging. The data support the main claims of the paper.

**Decision letter after peer review:**

Thank you for submitting your article "A Non-stop identity complex (NIC) supervises enterocyte identity and protects from pre-mature aging" for consideration by *eLife*. Your article has been reviewed by three peer reviewers, and the evaluation has been overseen by a Reviewing Editor and Jessica Tyler as the Senior Editor. The reviewers have opted to remain anonymous.

The reviewers have discussed the reviews with one another and the Reviewing Editor has drafted this decision to help you prepare a revised submission.

Neta-Erez et al., "A Non-stop identity complex (NIC) supervises enterocyte identity and protects from premature aging". The authors performed the RNAi screening on the maintenance of EC identity and identified the de-ubiquitinase Non-stop and its complex NIC as an EC identity supervisor. Using the similar techniques with their previous publication on *eLife* (Flint-Brodsly et al., 2019) and in combination with proteomic and genomic studies, the authors nicely demonstrated that the role of NIC on EC identity as well as its impact on aging. The data are convincing to support their claims but the following points need to be clarified to improve the manuscripts.

Essential revisions:

1) The relationship between non-stop (NIC) and other potential EC identifiers that are shown in Figure 1—figure supplement 1B-G is unclear. Since, these were not identified as part of the NIC complex, their contribution towards supervision of EC identity and whether they function downstream of Non-stop needs to be explained further. Based on the images in Figure 1—figure supplement 1, knockdown of CG1451, CG 18174 and CG1490 seems to have more delta positive cells than what is shown in Figure 1D, so the rationale behind characterizing Non-stop in more detail rather than other candidates and whether the role of other candidates overlaps with Non-stop is unclear. One question that comes to mind is whether increasing levels of CG1451, CG18174 or CG1490 can rescue the Non-stop phenotype.

2) The authors showed that Non-stop expresses basically all gut cell types, and EE-specific Not RNAi was no effect on EC identity. Hence, they mentioned "Non-stop function in maintaining gut identity was specifically localized in ECs". How about the effect of ISC or EB-specific Not RNAi on EC differentiation or identity? Loss of Non-stop in ECs results in reduced expression of 863 mRNAs with 38% of them being EC-related. How about the other 62%? Are they also important for cellular identity of other tissues? How many transcription factors are affected? Is Hey included? Are the NIC subunit genes transcriptionally downregulated upon Not(RNAi)?

3) The authors could show that Non-stop teams up with E(y)2, Sgf11, Cp190, mdg4, and Nup98 forming the Non-stop identity complex (NIC). However, direct binding was only confirmed for Cp190 in yeast 2H assays. Complex formation should also be shown for the other components. Depletion of Non-stop results in reduced Cp190, e(y)2, and Mod (mdg4) levels whereas Nup98 exhibits changed localization. The authors suggest that Non-stop stabilizes the associated NIC subunits but this model is not convincingly supported by the data. It should be ruled out whether the NIC subunit encoding genes are transcriptionally downregulated upon Non-stop depletion. Moreover, western blot analysis detecting the NIC subunits would be more quantitative to compare ECs (or S2R cells) with and without Non-stop expression. Otherwise it remains unclear whether the changes in NIC subunit level are directly resulting from Non-stop binding or not. CP190 binds directly to Non-stop via its 2nd/3rd zinc finger domain. Is a CP190 version lacking these binding domains more or less stable compared to wild-type CP190? It also remains unclear if NIC subunits are regulated ubiquitin-dependent. Could the NIC subunit level and/or Nup98 localization be increased by proteasomal inhibition? Rescue experiments with Non-stop mutants affected in ubiquitin-specific peptidase activity would be important to address the ubiquitin-dependent regulation mediated by Non-stop.

4) The authors showed mis-expression of Delta in the EC-specific Not RNAi flies in Figure 1 as one of the typical phenotypes for the loss of EC identity. However, they didn't show whether PPC** express Delta. The authors need to show this data or an appropriate explanation about this point is required.

5) Figure 1. The survival curve in Figure 1R is depicted for 33 flies only. For a reasonable inference to be drawn a minimum of 50-100 individuals should be included in a cohort if Log-rank test was used for statistical comparison. Also 3 repeats should be provided.

6) The Discussion is highly speculative and mainly focussed on supplementary data. It should be more focussed on the main findings of the work and provide a clear take home message with mechanistic concepts.

---

## [Author Response]

Essential revisions:1) The relationship between non-stop (NIC) and other potential EC identifiers that are shown in Figure 1—figure supplement 1B-G is unclear. Since, these were not identified as part of the NIC complex, their contribution towards supervision of EC identity and whether they function downstream of Non-stop needs to be explained further. Based on the images in Figure 1—figure supplement 1, knockdown of CG1451, CG 18174 and CG1490 seems to have more delta positive cells than what is shown in Figure 1D, so the rationale behind characterizing Non-stop in more detail rather than other candidates and whether the role of other candidates overlaps with Non-stop is unclear. One question that comes to mind is whether increasing levels of CG1451, CG18174 or CG1490 can rescue the Non-stop phenotype.

The aim of Figure 1—figure supplement 1B-G was to share with the readers examples of positive hits from the UB-identity screen. In G-TRACE analysis loss of Non-stop resulted in significant numbers of “EC cells that are no longer differentiated”. Moreover, at the same time we performed our Ub-screen we discovered via proteomic analysis the novel NIC complex. Therefore, we thought it will be interesting to test its potential role in cell identity specifically after that we realized that Nonstop activity is not mediated via SAGA complex, and that SAGA is not involved in regulation of EC identity. We do agree that other genes identified in the screen are highly interesting.

Taking into account the reviewer’s comment, we performed epistatic analysis between Non-stop and Hey is a master identity supervisor of EC identity. Specifically, we tested whether Non-stop expression suppresses Hey-related loss of identity phenotypes, and whether expression of Hey can suppress Nonstop LOF phenotypes. In both experiments no rescue was observed, suggesting for possible interdependent between Hey and non-stop. This notion is also supported with the observation that large number of EC genes that require both Hey or Non-stop for their expression. This new data is now shown in Figure 2—figure supplement 1E-H. and added the relevant section in the text:

“We previously identified genes that required Hey for their expression in ECs, and 76% (174/228) of Hey-dependent genes also required Non-stop for expression (Figure 5C). […] Likewise, expression of HA-Non-stop along elimination of Hey in ECs did not restore EC identity (Figure 2—figure supplement 1E-H)”.

As suggested by the reviewers an extensive epistatic analysis between genes identified in our screen and Non-stop and among themselves is interesting. However, towards this aim we are generating the appropriate UAS-transgenic lines and tools that at the current COVID-19 times are slowly progressing.

2) The authors showed that Non-stop expresses basically all gut cell types, and EE-specific Not RNAi was no effect on EC identity. Hence, they mentioned "Non-stop function in maintaining gut identity was specifically localized in ECs".

The reviewers are correct, Non-stop has a function in EC but not in EEs (the other differentiated cells in the midgut). However, it does have a role in progenitors (see below). Therefore, we corrected the sentence to:

“indicating that Non-stop function in maintaining differentiated identity was specifically to ECs and not EEs (Figure 1—figure supplement 2I-L).”

How about the effect of ISC or EB-specific Not RNAi on EC differentiation or identity?

We now show that Non-stop has a function in in progenitor biology. While this is not the center of the article we now show that loss of Non-stop results in hyperproliferation and mis-differentiation of progenitor cells. This new data is now shown in new Figure 1—figure supplement 2M-P and in the text:

*“*While we focused on differentiated midgut cells, we observed that Non-stop has also a role in the biology of progenitor cells. UAS-RNAi-mediated loss of Non-stop in progenitors (derived by the Esg>Gal4 deriver that is expressed in both ISCs and EBs), resulted in increased progenitor number (Delta positive small cells). […] However, a detailed analysis of the function of Non-stop in ISCs is outside the scope of this study that focuses on the differentiated ECs”.

Loss of Non-stop in ECs results in reduced expression of 863 mRNAs with 38% of them being EC-related. How about the other 62%? Are they also important for cellular identity of other tissues?

The reviewer’s comment is correct. We re-analyzed the data and corrected gene numbers in each group. As suggested we now present bioinformatic analyses of the genes that are not known EC genes and exhibit downregulated expression by Non-stop thus data is shown in figure. GO analysis unveiled that they are enriched in physiological pathways of ECs (new Figure 5—figure supplement 1A). We updated the text and figures:

“In addition, 51% (444/863) of Nonstop down regulated genes were not previously identified as EC genes in other studies. However, metascape analysis showed this group of genes is also enriched for EC-related pathways and physiology (Figure 5—figure supplement 1A).”

How many transcription factors are affected?

Overall we identified 13 transcription factors that require Non-stop for their expression, of these genes 7 TFs were factors that their expression is reduced upon expression of Esg in ECs. We present this new data in Figure 5—figure supplements 1A2, 1A3 and added the following text:

“Among Non-stop down-regulated genes were transcription factors including Odd-Skipped, and Relish, which is the central transcription factor of the IMD-innate immunity pathway that is involved in the local innate response of the gut to infection (Figure 5—figure supplements 1A2, 1A3)”.

Is Hey included?

No, based on our genomic analyses Hey is not regulated transcriptionally by Non-stop and Non-stop is not regulated transcriptionally by Hey. We added the following statement in the text:

“However, our expression data suggests that neither Non-stop nor Hey regulate the expression of one another (Figure 4—source data 1; Flint-Brodsly et al., 2019)”.

Are the NIC subunit genes transcriptionally downregulated upon Not(RNAi)?

Based on our RNA-seq analysis the expression of NIC subunits is not regulated by Nonstop at mRNA level. We added the following statement in the text:

“As evident by our RNA-seq analysis, the mRNA level of NIC subunits was not affected by loss of Nonstop in EC (Figure 5—source data 1).”

3) The authors could show that Non-stop teams up with E(y)2, Sgf11, Cp190, mdg4, and Nup98 forming the Non-stop identity complex (NIC). However, direct binding was only confirmed for Cp190 in yeast 2H assays. Complex formation should also be shown for the other components.

We further studied the interaction between members of NIC:

We now show that endogenous Non-stop interacts with endogenous Cp190 or Mdg4 in co immunoprecipitation assays (new Figure 3D). Using simple Y2H and in vitro binding assay we showed that Non-stop interacted directly with Ey2 and Cp190. Using Y2H we mapped the interaction within Cp190 (Figure 3E-G). Mdg4did not interact in vitro or in Y2H with Nonstop. However, via yeast 3 hybrid assay (Y3H) we learned that the interaction of Mdg4 requires the entire DUB module (Nonstop, Ey2,Sgf11), and mapped the region within Mdg4 that mediates the interaction.

All this new data is shown in new Figure 3G, and Figure 3—figure supplement 1F. We added this section in the text, and in addition clearly stated that additional experiments will be required to study the exact interactions within the complex.

“We further studied the interaction between Non-stop and members of the protein complex that we termed NIC (Non-stop identity complex), using endogenous coimmunoprecipitation from fly-derived protein extracts, in vitro binding assay, and yeast direct-hybrid systems (Figure 3D-H, Figure 3—figure supplement 1). […] However, further analysis will be required to fully establish the exact interactions interphases of proteins within the NIC complex”.

Depletion of Non-stop results in reduced Cp190, e(y)2, and Mod (mdg4) levels whereas Nup98 exhibits changed localization. The authors suggest that Non-stop stabilizes the associated NIC subunits but this model is not convincingly supported by the data.

Loss of Non-stop affects Nup98 resulting in two phenotypes: First, the overall level of Nup98 protein in ECs is reduced. Second, the localization of the remaining Nup98 is no longer confined to the nuclear periphery. We now present the data in a clearer way.

(see Figure 6K presenting separately localization from presence).

It should be ruled out whether the NIC subunit encoding genes are transcriptionally downregulated upon Non-stop depletion.

Based on our RNA-seq analysis the NIC subunits are not regulated by Non-stop at mRNA level. Thus, suggesting that the regulation of NIC subunits is only at the protein level. A relevant statement was added to the text.

Moreover, western blot analysis detecting the NIC subunits would be more quantitative to compare ECs (or S2R cells) with and without Non-stop expression. Otherwise it remains unclear whether the changes in NIC subunit level are directly resulting from Non-stop binding or not.

We performed the suggested and analyzed the levels of NIC subunit upon Nonstop depletion in EC, and using western blot analysis of extracts derived from these midguts. These experiments were not informative, as in whole midgut extracts, we did not detect reduction in the levels of Nonstop or other NIC subunits proteins. This is likely due to the robust increase in progenitor and other cells types that populate the midgut upon loss of Nonstop. These cells express Nonstop and NIC masking the reduction of Nonstop and NIC in ECs and ECs and EC-like cells that are observed visually by IF.

We have also performed similar experiments in S2 cells, where we eliminated Nonstop via shRNA. These preliminary experiments supported our midgut analysis in vivo and elimination of Nonstop resulted in reduced level of NIC subunits. However, these experiments were not fully reproducible technically and we decided not to include them in the current manuscript.

CP190 binds directly to Non-stop via its 2nd/3rd zinc finger domain. Is a CP190 version lacking these binding domains more or less stable compared to wild-type CP190? It also remains unclear if NIC subunits are regulated ubiquitin-dependent.

Due to the COVID situation we were not able to generate the reagent required for this experiment. The direct DUB activity of Nonstop is the center of a follow-up report that we are working on.

Could the NIC subunit level and/or Nup98 localization be increased by proteasomal inhibition?

To address this suggestion, we performed an *ex-vivo* experiment where midguts where Nonstop was eliminated in ECs for 12h. Subsequently, midguts were dissected, and the dissected midguts were incubated ex-vivo in the presence of increasing amount of the proteasome inhibitor MG-132 for 4h. However, we did not detect increase in the protein level of Non-stop, NIC subunits or improvement in the overall appearance of the midgut.We believe that future genetic analysis will be more appropriate for testing this question.

Rescue experiments with Non-stop mutants affected in ubiquitin-specific peptidase activity would be important to address the ubiquitin-dependent regulation mediated by Non-stop.

Unfortunately, we do not have such transgenic lines in hand, we are in the process of making such transgenes, that will be used to answer this question in the future.

4) The authors showed mis-expression of Delta in the EC-specific Not RNAi flies in Figure 1 as one of the typical phenotypes for the loss of EC identity. However, they didn't show whether PPC** express Delta. The authors need to show this data or an appropriate explanation about this point is required.

The reviewer is correct; We now added data showing Delta expression on the surface of PPC** as well as PPC*. This is now shown in Figure 2G, H and in the text:

“We observed a small number of PPC** that express the ISC marker Delta (<%1; Figure 2H, I). While this is a low percentage, we never observed the expression of Delta on the surface of control PPCs.”

“In addition, guts where Non-stop was targeted in ECs were populated with PPCs lacking expression of either RFP or GFP (termed PPC*), and are likely misdifferentiated progenitors that failed to activate the MyoIA-promoter and the entire RFP/GFP marking system, and ectopically expressed Delta (Figure 2H-J, Figure 2—source data 1)”.

5) Figure 1. The survival curve in Figure 1R is depicted for 33 flies only. For a reasonable inference to be drawn a minimum of 50-100 individuals should be included in a cohort if Log-rank test was used for statistical comparison. Also 3 repeats should be provided.

We have repeated the experiments and increased the number of individuals and our data are based on multiple biological repats. This new data is now shown in Figure 1R (LacZ control n=136; Nonstop RNAi n=149 p<0.0001).

6) The Discussion is highly speculative and mainly focussed on supplementary data. It should be more focussed on the main findings of the work and provide a clear take home message with mechanistic concepts.

We have extensively revised the Discussion and omitted the more speculative parts. We hope that now it has a better and concise take home message.